# The Aneto glacier's (Central Pyrenees) evolution from 1981 to 2022: ice loss observed from historic aerial image photogrammetry and remote sensing techniques

**Ixeia Vidaller[1], Eñaut Izagirre[2], Luis Mariano del Rio[3], Esteban Alonso-González[4], Francisco Rojas-Heredia[1], Enrique Serrano[5], Ana Moreno[1], Juan Ignacio López-Moreno[1], and Jesús Revuelto[1]**

[1]Instituto Pirenaico de Ecología, Consejo Superior de Investigaciones Científicas (IPE-CSIC), Saragossa CE1, Spain
[2]Department of Geography, Prehistory and Archaeology, University of the Basque Country UPV/EHU, Vitoria-Gasteiz, Spain
[3]Department of Applied Physics, Escuela Politécnica Superior de Cáceres, University of Extremadura, Cáceres, Spain
[4]Centre d'Etudes Spatiales de la Biosphère, Université de Toulouse,CNRS/CNES/IRD/INRA/UPS, Toulouse, France
[5]Department of Geography, GIR PANGEA, University of Valladolid, Valladolid, Spain

**Correspondence:** Ixeia Vidaller (ixeia@ipe.csic.es)

**Abstract.** TS1 The Aneto glacier CE2, although it may be considered a very small glacier ($< 0.5\,\mathrm{km}^2$), is the largest glacier in the Pyrenees. Its surface and thickness loss have been continuous in recent decades, and there have been signs of accelerated melting in recent years. In this study, thickness and surface losses of the Aneto glacier from 1981 to 2022 are investigated using historical aerial imagery, airborne lidar point clouds and CE3 unoccupied aerial vehicle (UAV) imagery. A ground-penetrating radar (GPR) survey conducted in 2020, combined with data from photogrammetric analyses, allowed us to reconstruct the current ice thickness and also the existing ice distribution in 1981 and 2011. Over the last 41 years, the total glacierised area has decreased by 64.7 %, and the ice thickness has decreased, on average, by 30.5 m. The mean remaining ice thickness in autumn 2022 was 11.9 m, as against the mean thickness of 32.9, 19.2 and 15.0 m reconstructed for 1981 and 2011 and observed in 2020, respectively. The results demonstrate the critical situation of the glacier, with an imminent segmentation into two smaller ice bodies and no evidence of an accumulation zone. We also found that the occurrence of an extremely hot and dry year, as observed in the 2021–2022 season, leads to a drastic degradation of the glacier, posing a high risk to the persistence of the Aneto glacier, a situation that could extend to the rest of the Pyrenean glaciers in a relatively short time.

## 1 Introduction

Glaciers are excellent indicators of climate variability and change because their evolution depends on the balance between snow accumulation during the cold period and ice/snow CE4 ablation during the warmest season (Braithwaite and Hughes, 2020). The Little Ice Age (LIA) represents the last cold pulse in almost all mountain ranges of the world (Solomina et al., 2016; García-Ruiz et al., 2020). As Grove (2004) and Oliva et al. (2018) point out, the LIA in the Pyrenees occurred during the period between the 14th and 19th centuries, in line with the rest of the Northern Hemisphere. Since $\sim 1850$, the LIA maximum, the climate has been warming and glaciers have been receding, albeit with brief periods of stabilisation or even small advances (Zemp et al., 2015; Oliva et al., 2018). However, the nearly continuous surface and thickness losses have accelerated in recent decades (Vidaller et al., 2021), similar to what has been observed in the majority of mountain ranges in the world (Hugonnet et al., 2021). The rapid surface and thickness losses are mainly due to a warming of more than $1.2\,°\mathrm{C}$ between 1949 and 2010 (Cuadrat et al., 2018), which could be even higher in high-elevation areas, affecting snow accumulation and its duration above ground (López-Moreno et al., 2019; López-Moreno, 2005). Due to the small size of Pyrenean glaciers, their evolution has strongly been influenced by the topographic characteristics of the surrounding area

(size and height of cirques, aspect, slope, snow avalanche corridors, etc.), and as well as having an interannual climatic control, they now also have a topoclimatic control (López-Moreno et al., 2006; Vidaller et al., 2021).

Consequently, the glacier surface loss in the Pyrenees is remarkable: there were 52 glaciers in 1850, 39 in 1984 and 21 in 2020, corresponding to an area of 2060 ha (20.6 km$^2$) in 1850, 810 ha (8.1 km$^2$) in 1984 and 232 ha (2.3 km$^2$) in 2020, representing a loss of 88.8 % of the glaciated area (Arenillas-Parra et al., 2008; Rico et al., 2017; Vidaller et al., 2021). In terms of ice thickness loss, unlike surface loss, there is generally a lack of information over a long period of time and a lack of sufficient resolution for small alpine glaciers (or very small glaciers). Recent studies have identified an ice thickness loss of 6.3 m for the period 2011–2020 as the mean for all the glaciers in the Pyrenean massif (Vidaller et al., 2021). Specifically, at the Monte Perdido glacier, López-Moreno et al. (2019) reported ice thickness loss of 6.1 m for the period 2011–2017. In the case of the Ossoue glacier, the ice thickness loss was 36.8 m for the period 1983–2013 and 20.4 m for the period 2001–2013 (Marti et al., 2015). In the grid cell corresponding to the Pyrenean glaciers (1° × 1° grids; 42° N, 0° E and 42° N, 1° W), Hugonnet et al. (2021) indicated a mean ice thinning rate of $-0.96$ m yr$^{-1}$ for the period 2000–2019, which is very accurate considering the dataset characteristics, but it is much higher than the mean annual ice thickness loss found by Vidaller et al. (2021) of $-0.70$ m yr$^{-1}$ for a more recent study period (higher ice loss could be expected in the later period). This difference between both studies clearly shows the need for local studies such as the present study or Vidaller et al. (2021) to validate large-scale observations and also to reach more accurate estimations over shorter time periods.

The Aneto glacier is one of the southernmost glaciers in Europe (Grunewald and Scheithauer, 2010) and is the largest in the Pyrenees, although it is a very small glacier (< 0.5 km$^2$) (Huss and Fischer, 2016). It is one of the most iconic glaciers of the Pyrenees, as it is located below the highest peak of the mountain range (Aneto peak, 3404 m above sea level (m a.s.l.)), and it forms part of the natural and cultural landscape of the Posets–Maladeta Natural Park, attracting mountaineers and tourists to this park (Carvache-Franco et al., 2022; Carrascosa-López et al., 2021). Additionally, this glacier is part of the Natural Monument of the Pyrenean Glaciers (Lampre-Vitaller, 2003), adding additional societal value to this natural landscape heritage CE5. Unlike other alpine glaciers that are important water sources in other mountain areas (Fountain and Tangborn, 1985; Braithwaite and Raper, 2002; Meier et al., 2007; Huss et al., 2017; Drenkhan et al., 2023), the Aneto glacier, as all Pyrenean glaciers, has a minor (and nearly negligible) contribution to river discharge in this region (López-Moreno et al., 2020). However, the ice surface loss of Pyrenean glaciers has a clear impact on local erosion rates (Riihimaki et al., 2005), nutrient fluxes, biochemistry and macroinvertebrate communi-

ties (Snook and Milner, 2001; Brown et al., 2007) or the microbiology of these emblematic landscapes and surrounding downstream areas. The knowledge gap of these processes in the southernmost glaciers of Europe encourages and justifies the analysis of their recent evolution.

Despite the fact that the Aneto glacier has not been subjected to mass balance annual monitoring, two recent studies (Campos et al., 2021; Vidaller et al., 2021) have analysed ice thickness loss for different time periods. Campos et al. (2021) presented a reconstruction of the area, volume, ice thickness and equilibrium line altitude (ELA) of the Aneto glacier for different time periods from the LIA to 2017 using photo interpretations and satellite imagery to calculate surface and ice thickness losses in the Aneto glacier. Ice thickness loss in that work was derived from a steady-state model assuming a plastic ice rheology, combined with some ground-penetrating radar (GPR) profiles from 2008 (Campos et al., 2021). On the other hand, Vidaller et al. (2021) determined changes in glacier area and thickness for the period 2011–2020 with high spatial resolution in the 24 Pyrenean glaciers (including the Aneto glacier). Surface loss was determined based on satellite data and drone imagery, and the ice thickness loss was calculated by comparing 2011 and 2020 digital elevation models (DEMs) (from laser imaging detection and ranging (lidar) and unoccupied aerial vehicles (UAVs), respectively). The results of this work for the Aneto glacier reported a surface loss of 24.9 % (69.3 ha (0.7 km$^2$) in 2011 and 50.0 ha (0.5 km$^2$) in 2020) and an average ice thickness loss of 8.5 m.

This study aims at analysing the recent evolution of the highest and largest glacier of the Pyrenees, the Aneto glacier, by using the longest temporal dataset of glacier thickness loss in the Pyrenees. In addition, this study permits us to assess the impact of a single extremely warm ablation season (2022) on glacier evolution. Due to the very last stage in which the Aneto glacier is, we report thickness and ice surface losses of this glacier from 1981 to 2022 to discern if the speed of changes accelerates (because of the existence of feedback processes) or slows down (because the remaining ice is progressively restricted to the most favourable areas), which has an inherent scientific interest and may be extrapolated to other mountain areas that will face a similar situation in the coming decades. The evidence for the demise of Pyrenean glaciers in the coming decades using the Aneto glacier as an iconic example is also used to highlight the dramatic consequences of rapid climate change in mountain areas CE6. We use high-resolution 3D point clouds from 1981 (from structure-from-motion (SfM) methods exploiting historical aerial photographs), 2011 (from the Spanish National Geographic Institute (IGN) lidar survey), 2020, 2021, and 2022 (from SfM methods using UAV flights). In addition, 2020 ice thickness was estimated from an intensive GPR survey conducted in July of this year. The combination of the three techniques allows for the accurate reconstruction of the glacier ice thickness in 1981 and its evolution until today. Moreover, the current ice thickness and basal topography of the glacier

could be determined. This information is critical for predicting the next changes in the glacier, and the basal topography reveals sectors where lake formation is likely after the ice disappears. The combination of these techniques provides an increase in knowledge over previous work because (1) we present data with high accuracy and lower uncertainty compared to previous studies, and (2) we determine the evolution of the Aneto glacier for the longest period observed by quantifying current ice thickness and basal topography, as well as the annual decrease in ice thickness from 1981 to 2022.

## Study area

The Aneto glacier is the largest glacier in the Pyrenees (48.1 ha ($0.48 km^2$) in 2022), a mountain range where only four glaciers are larger than 10 ha. It is located in the Maladeta massif (Fig. 1), on the northeast (NE) side, between the Maldito (3354 m a.s.l.) and Aneto (3404 m a.s.l.) peaks. The high elevation of this massif, with more than 40 peaks above 3000 m a.s.l., has allowed the preservation of other smaller glaciers (eastern Maladeta and Tempestades) and ice patches (western Maladeta, Coronas and Barrancs) in the area. In 2022, the Aneto glacier consisted of two bodies whose glacier front was at 3026 m a.s.l. in the case of the main body and at 3170 m a.s.l. in the case of the secondary body.

In this area, the 0 °C mean annual isotherm ranges from 2700 to 3000 m a.s.l. (Jomelli et al., 2020), and the mean annual precipitation is about 2000 mm, with winter and spring being the wettest seasons (Buisan et al., 2015). The mean annual temperature for the period 2007–2022 was 4.6 °C at the weather station of the Renclusa hut (2140 m a.s.l.); meanwhile the mean temperature for the same period in the ablation season (June–September) was 11.6 °C. The year 2022 was an especially warm year, in which the annual mean temperature was 5.2 °C and the summer mean temperature was 12.1 °C (data from the AEMET database).

## 2 Data and methods

### 2.1 Imagery processing and DEM generation

#### 2.1.1 Historical aerial imagery

The earliest imagery dataset exploited here (1981 DEM) dates from September 1981. Aerial images were acquired by the Spanish National Geographic Institute (IGN) using analogue photogrammetric cameras (IGN: http://centrodedescargas.cnig.es/CentroDescargas/index.jsp, last access: August 2022) aboard aircraft for national mapping surveys. The objective was to collect aerial photographs suitable to produce topographic maps of Spain at a scale of 1 : 50 000 and with contour intervals of 20 m (named MTN50). The overlap was 60 % at the front and 30 % on the side. The camera, Wild lens cone RC 10, had a sensor of $230 \times 230$ mm,

a lens of 15 UAG II and a focal length of 152.12 mm; thus, an average image scale of 1 : 30 000 was obtained, with a ground sampling distance (GSD) between 0.35 and 0.18 m per pixel. For this study, the historical aerial imagery was rescanned at a resolution of 15 μm. A total of 18 aerial images of the Aneto massif were used, taken from the same flight in late summer 1981.

Historical survey imagery was processed using structure-from-motion (SfM) (Snavely et al., 2006) with Agisoft Metashape Professional v1.6.3 software (https://www.agisoft.com/, last access: TS2), which has shown reliable results when used for processing historical images (Llena et al., 2018). Processing parameters were set according to official Agisoft guidelines (denser point clouds, bundle block adjustment (BBA), internal and external camera parameter calibration; Agisoft Metashape version 1.5, 2019). The SfM routines enabled the generation of a dense point cloud ($2.4 pts m^{-3}$ TS3), from which an orthomosaic with a resolution of 0.41 m (used to calculate the glacier area) and a geoid-corrected digital terrain model (DTM) with a grid cell size of 1.58 m were derived.

The historical survey imagery processing included the following workflow: (1) the alignment of each flight line's cameras (three lines in total); (2) the assignment of ground control points (GCPs) based on clearly visible features such as individual large boulders and trail crossings or mountain summits; (3) the derivation of accurate geographic coordinates and elevation information of these later GCPs using high-resolution satellite imagery (DigitalGlobe/GeoEye-1 imagery with 1 m resolution available through the QGIS service QuickMapServices) and a 2020 UAV flight as a reference DTM (Vidaller et al., 2021); and, (4) taking advantage of GCPs, the realignment of camera positions and merging of all images in one chunk using Agisoft Metashape Professional. The georeferencing accuracy of DigitalGlobe's latest very-high-resolution (VHR) satellites (i.e. GeoEye-1 and WorldView-1/2/3/4) is between 1.0 and 5.0 m, which may be insufficient for many precise geodetic applications. To improve this, we aligned the 1981 point cloud with that of 2020 using the ICP CE7 algorithm (Rajendra et al., 2014).

#### 2.1.2 Lidar survey

The 2011 high-resolution digital elevation model (DEM) was derived from airborne lidar. The data were acquired in a flight of 9 November 2011 by the IGN (http://centrodedescargas.cnig.es/CentroDescargas/index.jsp, last access: May 2022). The lidar device was the Leica ALS60 with a diode-pumped transmitter and a low-inertia/high-speed scanning mirror with a large aperture operating at a wavelength of 1064 nm. The final georeferenced point cloud had an average density of $0.35 pts m^{-3}$. This information was processed and accurately geolocated by the IGN, which provides free access to the final 3D point cloud.

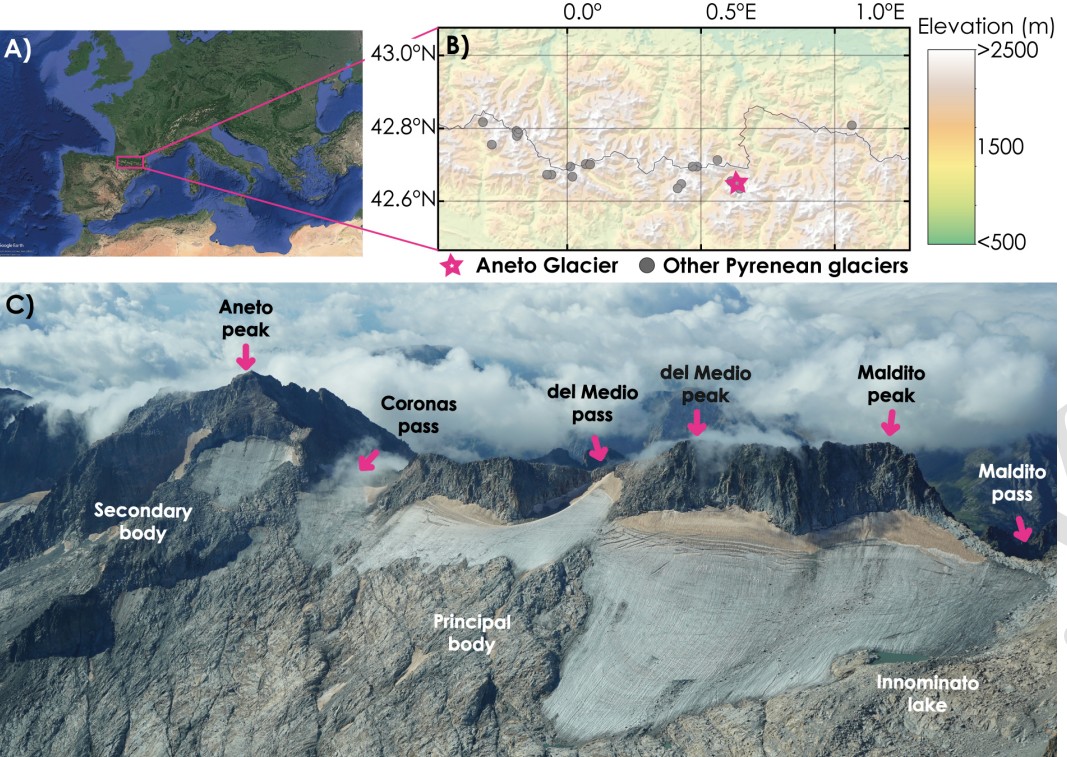

**Figure 1.** Location of the Aneto glacier. **(a)** Map of Europe, with the pink rectangle delimiting the central part of the Pyrenees (© Google Maps). **(b)** Topographic map of the central Pyrenees; the glaciers in this area are marked with grey dots, and the location of the Aneto glacier is marked with a pink star. **(c)** An aerial photo of the Aneto glacier in summer 2021. The main reliefs surrounding the glacier are indicated.

### 2.1.3 Unoccupied aerial vehicle (UAV) imagery

The 2020, 2021 and 2022 glacier surface DEMs were obtained using a fixed-wing UAV (SenseFly eBee X) on 12 September 2020, 1 October 2021 and 10 September 2022, respectively. The UAV was equipped with a SenseFly 3D S.O.D.A. digital camera (20 Mp resolution) and GPS receivers enabling post-processed kinematic (PPK) positioning systems (positioning accuracy < 0.05 m after post-processing). As in previous studies (e.g. Vidaller et al., 2021), the UAV images had an overlap of 70 % at the front and 50 % on the side (note that the 3D S.O.D.A. camera obtains images with a tilt of 30°) with a final ground sampling distance (GSD) of 2.8 cm per pixel. The UAV images were processed using Pix4Dmapper (Pix4D) SfM software, in which the calculation of BBA and internal and external camera parameter calibration were enabled (more details on data processing can be found in Vidaller et al., 2021). Although Agisoft Metashape could be used for this SfM processing, we preferred to use the same protocol described in previous works with UAV at this site. Nonetheless comparison of point clouds from the SfM software (both Pix4Dmapper and Agisoft Metashape) shows equivalent accuracies to work in this area (Mölg and Bolch, 2017; Llena et al., 2020). Due to the three UAV acquisitions having the same acquisition proto-

col, and the GPS PPK geolocation (image geolocation with deviations below 4 cm), the comparison of these three point clouds yielded negligible deviations (0.06 m) (Revuelto et al., 2021).

## 2.2 In situ ground-penetrating radar (GPR), processing and data interpolation

GPR uses the transmission and rebound of electromagnetic pulses at different frequencies to determine ice thickness and glacier interfaces (rocks, bedrock basin, snow, etc.) (del Rio et al., 2014). Different works have studied the variation in ice thickness, surface area or volume on glaciers using different techniques, which highlight the importance of the methodology to be applied in each case, considering its scope and limitations (Procházková, 2019; Bohleber et al., 2017; Marcer et al., 2017; Fischer, 2009 TS4).

GPR fieldwork was conducted on 25–26 July 2020, using a Malå Geoscience radar system consisting of a ProEx control unit and a 100 MHz rough terrain antenna (RTA). Occasionally several transects were also carried out with the 100 MHz shielded antenna (see Supplement). Georeferenced radargrams were created using the AtlasLink GNSS smart GPS antenna connected to the GPR, which were obtained in "time" tuning. A total of 32 georeferenced radargrams

were recorded in the main glacier body in a common off-set mode, corresponding to a length of 6.8 km and covering almost the entire glacier surface (more detailed information can be found in Fig. S1 in the Supplement). The campaign was conducted during a period when the glacier surface was covered with snow, in order to allow safe displacement of the instrument and operators, thus hampering the observation of deeper ice layers. This required differentiation of the snow layer in post-processing to accurately quantify glacier thickness.

Radargrams were processed using Reflexw version 9.1.3 (Sandmeier scientific software), with the following workflow: (1) the adjustment of the time origin ($t = 0$) to coincide with the arrival of the first surface signal on the glacier; (2) the homogenisation of the trace increment, since the acquisition of the radargrams with the RTA antenna was done in time mode and varied in each radargram depending on the speed of the movement of the antenna on the ground ($0.1 \, \mathrm{m \, ns^{-1}}$ was fixed, since this was the smallest value obtained in the radargrams); (3) the removal of the background; (4) the correction of the energy loss of the signal when penetrating the terrain by applying a gain factor of 0.2 (energy decay); and (5) the application of a frequency bandpass filter so that only signals with frequencies between 50 and 200 MHz remain (the nominal frequency of the antenna is 100 MHz).

As a first approximation, $0.17 \, \mathrm{m \, ns^{-1}}$ was set as the propagation velocity of the waves in the glacier to get a first idea of the thickness of the snow and ice layers in the radargram representation. Snow and ice layers must be defined from the radargrams to create a thickness model of both. To do this, the wave propagation velocities (RWVs) in both media must be available beforehand. In a similar study on the Monte Perdido glacier, RWVs of $0.200 \pm 0.005 \, \mathrm{m \, ns^{-1}}$ for snow and $0.163 \pm 0.007 \, \mathrm{m \, ns^{-1}}$ for ice were obtained for the 500 and 200 MHz antennas, respectively (López Moreno et al., 2019). The coherence of these velocities was checked in the 1054 radargram at the points where diffraction hyperbolas occurred (plot of diffraction hyperbolas is shown in Fig. S2).

The distribution of the GPR data does not follow a homogeneous pattern; the GPR record tracks were distributed along parallel and perpendicular lines, forming an irregular grid (Fig. S1). Therefore, to determine the thickness of the glacier over its entire extent, an interpolation method is required. For this type of data, the interpolation method used was the radial basis function (RBF), as Otero-García (2008) recommended. Given the poor distribution of the data, after several tests, the best method is to work with 16 neighbours, two per octant (the closest points in each direction), in a circular area with a radius of 457.62 m, in the same way, again, as Otero-García (2008). The thickness for glacier limits in 2020 was established as 0 m. To validate this interpolation method, the data were divided into two groups: training with 70 % of the sample and test with the other 30 %.

## 2.3 Glacier area outline, point cloud geolocation and glacier thickness loss computation

The delineation of the Aneto glacier surface was done manually (Table S4 in the Supplement) in a GIS software (ArcGIS), considering: (1) the orthomosaic of the historical aerial imagery from 1981; (2) a RapidEye satellite image from 2011 and improved outlines from RGI (RGI Consortium, 2017 TS5); and (3) the orthomosaics derived from UAV flights in 2020, 2021, and 2022. Due to the small extent of these very small glaciers, the slope was considered in the calculation of glacier surface to obtain the true glacier area (3D surface) rather than the 2D projection of glacier extent. This calculation is justified because the glaciers are strongly bound to wall cirques, and these had a steepness of 24.3° in 2020. When the slope is not taken into account, the glacier surface is underestimated (Vidaller et al., 2021). Otherwise the 2D area computation would also be affected by the changes in slope during the study period.

Data from DEMs available for this work varied in accuracy. The most accurate geolocation is that of the UAV, which was used as a reference for the point cloud due to the post-processed kinematic (PPK) GPS geolocation technique (geolocation RMSE < 0.05 m). This geolocation error is equivalent for the 2020, 2021 and 2022 point clouds (0.019 for 2020, 0.025 for 2021 and 0.021 for 2022; the differences were due to weather conditions). Based on the low magnitude of these geolocation errors, we assume that the error introduced in ice thickness differences is nearly negligible. 3D point cloud differences in ice-free areas had RMSEs below 0.02 m, (error computed following Vidaller et al.'s (2021) accuracy method). To coregister the lidar point cloud (2011) and the point cloud from the historical aerial imagery (1981), several areas of stable terrain such as ridges, peaks, polished surfaces, etc. were selected in these later point clouds and in the 2020 UAV-derived point cloud. These areas were evenly distributed around the glacier. A rotation and translation matrix was calculated for these areas to align (separately) the 1981 and 2011 point clouds with that of 2020 using an ICP algorithm (Rajendra et al., 2014), from CloudCompare software (Girardeau-Montaut, 2016), in the same way as Vidaller et al. (2021). Subsequently, these matrices were applied to the entire point clouds to derive point clouds that were finally coregistered. Glacier thickness loss (normal surface differences; see the Supplement for more information) between these point clouds were computed using the CloudCompare tool M3C2 (James, 2017) to determine the differences (surface perpendicular) between the glacier surfaces observed in different years. Glacier change statistics were derived from this later comparison, calculated over the most recent (and smallest) glacier surface.

Glacier thickness loss was determined by considering only data within the smallest (or more recent) surface of the glacier. When considering the oldest surface, there are zones of the glacier that are not present in the most recent acqui-

sitions, so the ice thickness loss would be underestimated (Vidaller et al., 2021). The mass balance was calculated assuming a density conversion factor of $850 \pm 60$ kg m$^{-3}$ (Huss, 2013). Thus, the specific mass balance presented in this study was determined considering the recent surface of the glacier.

With the aim of determining areas of future glacier lake formation, the mountain basal topography was derived from the GPR interpolation and the 2020 UAV acquisition (subtraction of the 2020 glacier surface from the ice thickness interpolation from the GPR). The topographic position index (TPI) is capable of identifying terrain depressions at various search distances (Weiss et al., 2001). From this basal topography, the TPI (de Reu et al., 2013) was derived for 70, 100, 150 and 200 m search distances to describe depression areas that potentially favour future lake formation. This index has previously been used in studies of debris-covered glaciers (Westoby et al., 2020) to determine areas of potential debris accumulation, but as far as the authors are aware, this is the first time this index has been used to determine areas of potential lake formation following the retreat of mountain glaciers. In addition, overdeepenings detected by the TPI were corroborated using the longitudinal GPR radargrams.

## 2.4 Correction and accuracy assessment

GPR ice thickness measurements with a 100 MHz RTA antenna are subject to intrinsic error. Assuming a RWV velocity for ice of 0.163 m ns$^{-1}$, the $\lambda$ value is 1.63 m, so the minimum spatial resolution is $\lambda/2 = 0.815$ m. Summing this uncertainty for snow and ice gives a thickness resolution of 1 m for this delineation. Thus, the uncertainty in the determination of the ice layer thickness is 1.8 m.

To check the coherence of the determined thicknesses, a test was performed at all intersections between transects to detect any inconsistencies in the values. At these 28 intersections, the average difference is $1.6 \pm 1.6$ ($\sigma$) m, with some outliers of 3–5 m (Table S2). This value is consistent with the uncertainty associated with RWV velocity and ice layers' delineation (1.8 m). The lengths of the radargrams were determined using Reflexw from the GPS coordinates coupled to the GPR (see Supplement for more details). General GPR uncertainty in ice thickness was determined considering different velocities for temperate ice in the transects (1043, 1062 and 1073). Based on existing literature (Jimenez, 2016 TS6; López-Moreno et al., 2018 TS7), we assumed 0.2 m ns$^{-1}$ in the snow and between 0.157 and 0.186 m ns$^{-1}$ in the ice. With these velocities, mean and maximum ice thickness was determined for each transect (Table S3). As a result, mean ice thickness variation that could be derived from different velocities into the temperate ice would fit in the range of the estimated margin of error band ($< 1.6$ m) and would be smaller than the uncertainties obtained from the differences in thickness at transect crossings ($< 1.8$ m).

To validate the interpolation of glacier thicknesses, 30 % of the GPR data were randomly selected, and the remaining 70 % of the GPR dataset was used for the interpolation (Otero-García, 2008). The mean error between the interpolated thickness and the thickness observed with the GPR was 0.0018 m, and the RMSE was 0.3021 m.

The delineation of glacier boundaries also has some uncertainty due to pixel size, geometric correction, visual identification, and the presence of residual snow or debris cover at the glacier boundaries. The surface uncertainty is 0.048 ha (0.00048 km$^2$) for the Aneto glacier (Vidaller et al., 2021) in the case of the glacier surface of 2011, 2020, 2021 and 2022; the uncertainty error of the 1981 glacier outline is 0.58 ha (0.0058 km$^2$).

The coregistration of point clouds from historical aerial imagery and lidar survey with UAV surveys was tested in a buffer zone around the glacier, always using snow- and ice-free zones in both years of comparison. This means that the comparison of the 1981 and 2020 point clouds was performed in a buffer zone with a 300 % larger extent than the 1981 glacier boundaries (over stable terrain); the coregistration error between the 2011 and 2020 point clouds was determined in the same way. In the first case for the Aneto glacier the RMSE is 0.06 m and in the second case 0.4 m (Vidaller et al., 2021).

## 3 Results

The extent of the Aneto glacier has decreased significantly in the last few decades, from 135.7 ha (1.36 km$^2$) in 1981 to 48.1 ha (0.48 km$^2$) in 2022, i.e. by $-64.7$ %. The surface and thickness losses of the glacier continues, resulting in changes in area and the division of the glacier into two bodies. It is noteworthy that the secondary body today shows signs of stagnant dynamics (Table S5).

In 1981, the surface of the Aneto glacier was 135.7 ha (1.36 km$^2$); in 2011, the surface decreased to 69.3 ha (0,69 km$^2$), a loss of 49.0 %. Between 2015 and 2016, the Aneto glacier divided into two bodies; in 2020 the main body was 47.8 ha (0.48 km$^2$) and the secondary body was 4.2 ha (0.04 km$^2$), a total of 52.0 ha (0.52 km$^2$). Table S5 shows that in the last 40 years the losses were 63.1 % of its surface ($-1.6$ % yr$^{-1}$). In 2022, the surface had decreased to 48.1 ha (0.48 km$^2$) (44.6 ha (0.45 km$^2$) for the main body and 3.52 ha (0.03 km$^2$) for the secondary body), a decrease of 64.7 % compared to 1981 (Fig. 2). This decrease represents a retreat of the lowest glacier front (the front of the main body) from 2828 m a.s.l. in 1981 to 2939 m a.s.l. in 2011, 3011 m a.s.l. in 2020, 3014 m a.s.l. in 2021 and 3026 m a.s.l. in 2022.

A comparison of the 1981 and 2022 point clouds (difference calculated normal to surface) shows a mean ice thickness loss of 30.51 m (Figs. 3, S3 and S4) in this period and considering only the area occupied by the glacier in 2022 CE8 (considering the 1981 glacier extent, the ice thickness loss is 24.1 m; and considering height surface changes the losses are 45.3 m (for more information see Table S6 and Fig. S4)).

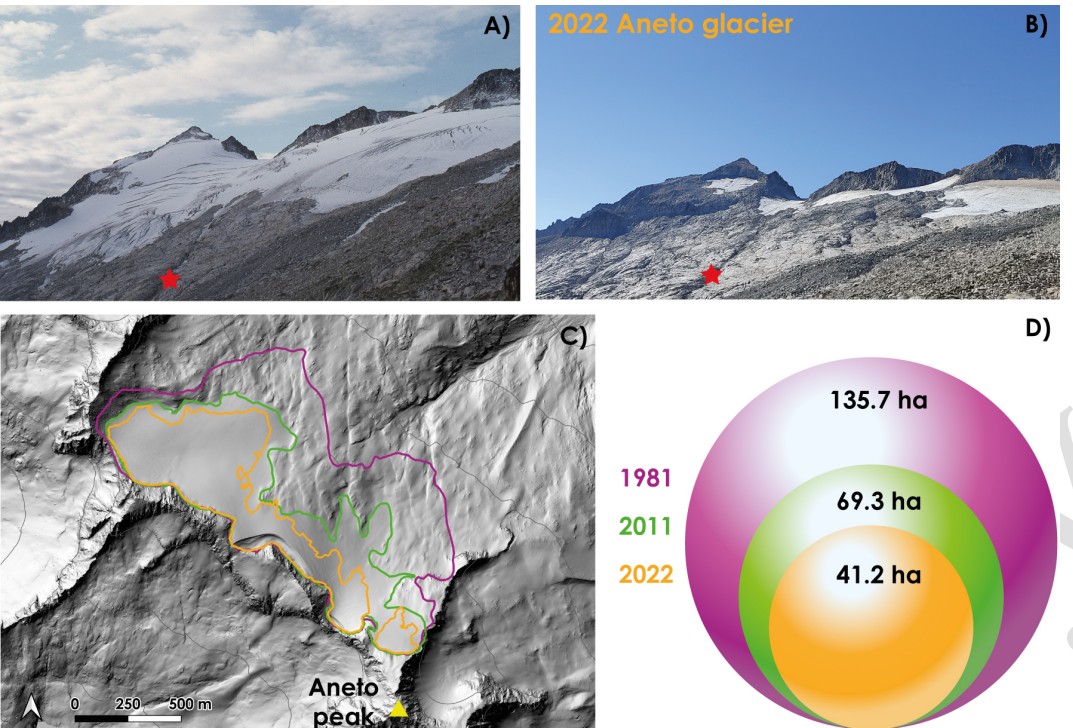

**Figure 2.** Appearance of the Aneto glacier during the study period. **(a)** Photo (Fernando Biarge, Fototeca DPH) corresponding to the Aneto glacier in 1982. **(b)** Photo corresponding to the Aneto glacier in 2022. The red stars refer to the same location in both photos. **(c)** Map showing the differences in the area of the glacier during the study period; the purple line delineates the extent of the glacier in 1981, the green line in 2011 and the orange line in 2020. The shading of the terrain was calculated from the 2011 lidar. The yellow triangle represents the summit of the Aneto peak. **(d)** Cumulative area change plot of the Aneto glacier for the years 1981 (purple), 2011 (green) and 2022 (orange).

Note these mean ice thickness losses are the mean values of differences in glacier surfaces (normally computed) for the entire period computed. This means that the glacier lost on average $0.6 \, \text{m yr}^{-1}$ over the entire glacier and $0.7 \, \text{m yr}^{-1}$ in the currently glaciated area during the 1981–2022 period. The thickness losses are not evenly distributed. The highest ice thickness loss is in the middle of the main body, while the lowest changes are in the secondary body (Fig. 3a). More than 41 % of the 2022 glacier area has lost more than the mean (30.5 m) (Fig. 3b).

The results indicate an acceleration in glacier ice thickness loss in the last decade. The mean ice thickness loss for the period 1981–2011 was 17.8 m ($0.6 \, \text{m yr}^{-1}$) and 12.6 m ($1.1 \, \text{m yr}^{-1}$) for the period 2011–2022, representing an increase in ice thickness loss in the later period of 200 % compared to 1981–2011. The available information for the 2020–2021 and 2021–2022 annual comparisons highlights the high interannual variability in ice thickness loss, with mean ice thickness loss of 1.5 and 3.2 m, respectively. As for the specific mass balance, the changes are $-0.6 \, \text{m w.e. yr}^{-1}$ for the period 1981–2022, $-0.5 \, \text{m w.e. yr}^{-1}$ for the period 1981–2011, $-1.0 \, \text{m w.e. yr}^{-1}$ for the period 2011–2022, $-1.2 \, \text{m w.e. yr}^{-1}$ for the period 2020–2021 and $-2.7 \, \text{m w.e. yr}^{-1}$ for the period 2021–2022 (data are always calculated within the most recent glacier surface).

The GPR survey of the main body of the glacier in 2020 reveals a mean glacier thickness of 15.0 m, with a maximum glacier thickness of 44.7 m (Fig. 4a). This maximum glacier thickness was measured in the western part of the glacier, near the Maldito (3354 m a.s.l.) and del Medio (3349 m a.s.l.) peaks. The greatest thickness was measured in the upper parts of the glaciers in the elevation range between 3200 and 3350 m a.s.l. (Fig. 4a and b). In some elevation ranges (between 3100 and 3180 m a.s.l.), the glacier thickness is lower than expected, considering the trend of increase with increasing elevation. This is mainly due to the presence of a relatively thick sector (up to 39 m) between 3000 and 3100 m in the western part of the glacier, which affects the mean values observed in this elevation range. Figure 4a also shows the presence of very narrow and shallow ice sectors (light blue areas) adjacent to the cirque wall in two places, indicating an imminent separation of the glacier into three ice bodies.

In 1981 (Fig. 5a), the pattern of ice thickness distribution shows some differences compared to recent periods. In 1981, the maximum glacier thickness was found in the middle elevations of the western part, where ice thickness reached 90 m. Below the del Medio pass, the glacier thickness was also very

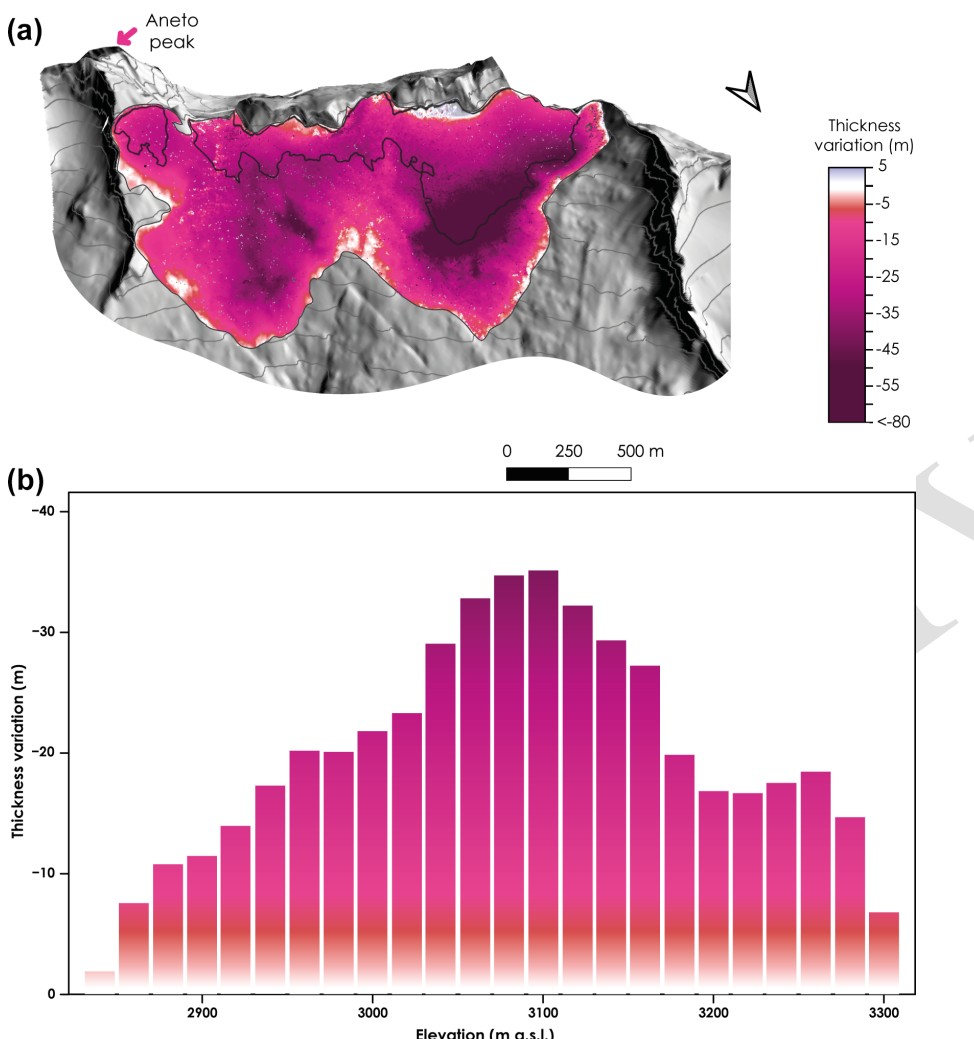

**Figure 3. (a)** Thickness loss of the Aneto glacier from 1981 to 2022. In the upper map, the black line delineates the glacier in 2022, while the grey line represents the glacier in 1981. The arrow indicates the north direction (see the maps in Fig. S3 for each period of the UAV surveys). **(b)** Distribution of thickness loss considering elevation bands (mean of each band) of 20 m.

thick, almost 70 m. In 1981, the maximum thickness was 96.5 m, and the mean thickness of the glacier was 32.9 m. In 2011 (Fig. 5b), the distribution pattern of ice thickness on the Aneto glacier was very similar to that of 2020 (Fig. 4a); the maximum ice thickness was measured below the Maldito peak and in the lower western part of the glacier. The maximum ice thickness at that time was 52.5 m, while the mean ice thickness of the glacier was 19.2 m. In 2022 (Fig. 5c), the ice thickness distribution had not changed markedly, and the greatest thickness was also under the Maldito pass and peak, as well as in the middle of the main body of the Aneto glacier. In this latter year, the average ice thickness was 11.9 m, and the maximum ice thickness was 44.0 m, but although the maximum ice thickness exceeded 44 m, 43.0 % of the Aneto glacier in 2022 had an ice thickness of less than 10 m.

Glaciers erode the surface beneath the ice mass so that the subglacial topography is not a flat surface (Palacios et al., 2020 TS8). Glacial erosion creates thresholds and depressions, which in some cases are filled by meltwater from the glacier, forming glacial lakes (Shugar et al., 2020; Yao et al., 2018). This is the case with Ibón Innominato, a new, small proglacial lake formed in 2015 as a result of the retreat of the Aneto glacier. Today, it is the highest mountain lake in the Pyrenees (3150 m a.s.l.). Due to the continuous surface loss of the glacier, this lake grows simultaneously with the retreat of the Aneto glacier, although it is ice free only 3–4 months a year (July–October). In 2020, its area was 0.4 ha and in 2022 it was 0.5 ha, an increase in area of 26.5 % for the period 2020–2022, mainly due to the frontal retreat of the Aneto glacier by about 15.2 m.

The TPI spatial distribution depicts depression areas that could fill with water after the ice disappears (blue colours in Fig. 6). For example, under the del Medio peak and Medio pass a remarkable depression for CE9 150 and 200 m search

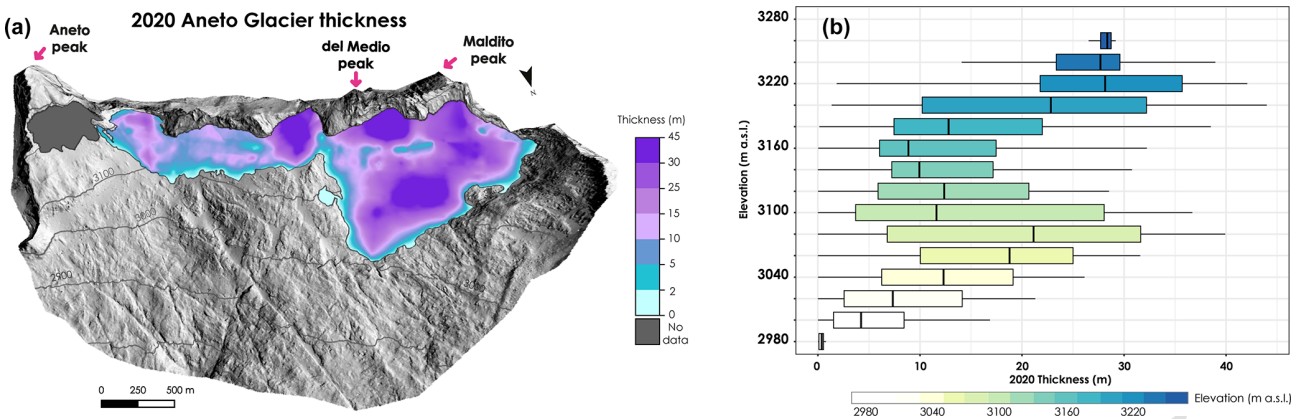

**Figure 4.** Ice thickness of the Aneto glacier in 2020. In map **(a)**, the blue colour represents the zones of lesser ice thickness that are about to disappear, in contrast to the purple colours that represent the greatest ice thickness. The secondary body of the Aneto glacier is coloured grey because no data are available for this glacier body, and therefore no interpolation is possible. The boxplot **(b)** shows the mean glacier thickness in 2020 for each elevation band (20 m). A GPR profile is shown in the Supplement as an example of the longitudinal radargram (SE–NW) of the glacier (Fig. S5).

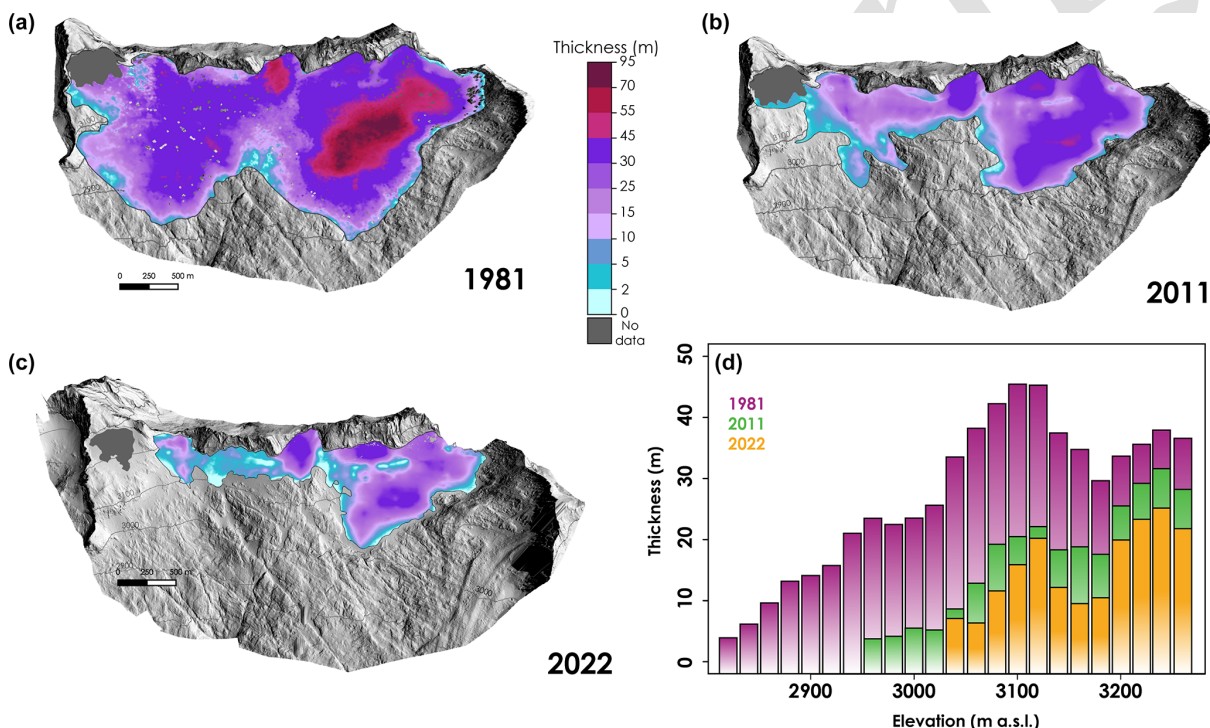

**Figure 5.** Reconstruction of the ice thickness of the Aneto glacier at different times during the study period. Panel **(a)** shows the thickness in 1981, **(b)** shows the thickness in 2011 and **(c)** shows the thickness in 2022. The blue colour represents the zones of lower ice thickness that are about to disappear, in contrast to the red colours that represent the greatest ice thickness. The secondary body of the Aneto glacier is coloured grey because no data are available for this glacier body, and therefore no interpolation is possible. **(d)** Comparison of the thickness of the Aneto glacier in 1981, 2011 and 2022, with structures in elevation bands of 20 m.

distances is observed. This spatial distribution of the lowest value of TPI is confirmed by radargram 1062 (Fig. S5), in which the left side coincides with the overdeepening area below the del Medio pass and also with the second depression below the Maldito peak. These areas nowadays have

the highest ice thicknesses, and thus lakes could be found in these areas when the glacier has completely disappeared.

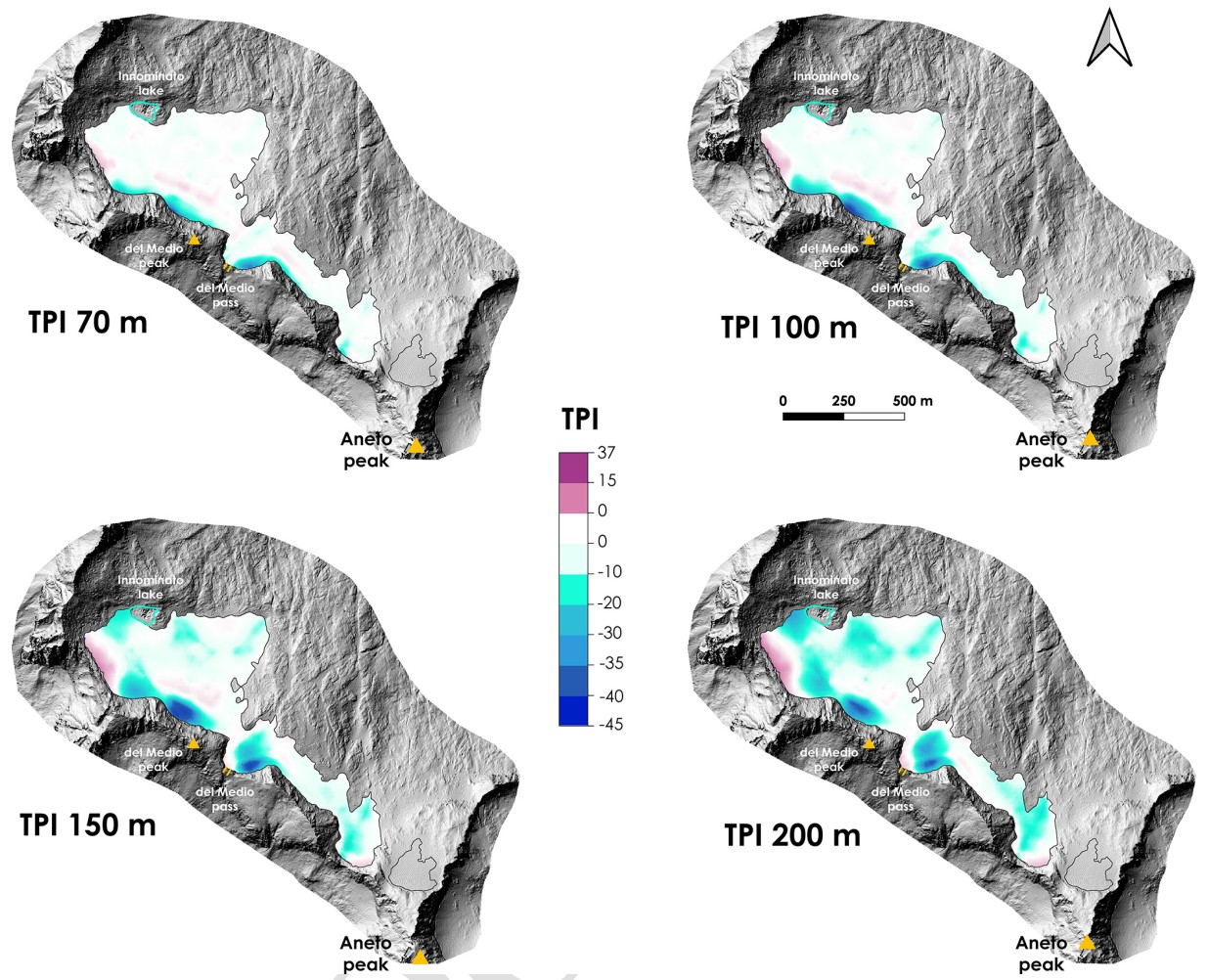

**Figure 6.** TPI 70, 100, 150 and 200 m based on the basal topography derived from the GPR data from 2020. Negative (positive) values (blue (red) colours) represent locations that are lower (higher) than their surroundings.

## 4   Discussion

### 4.1   Recent changes in the Aneto glacier: a foreshadowing of the future evolution of European glaciers

Annual surface loss has decreased uniformly over time ($-2.2\,\mathrm{ha\,yr^{-1}}$). However, it must be noted that the relative changes are larger in the latter years, since the losses occurring in the most recent period are measured with respect to a progressively smaller surface. Thus, there has been no recent acceleration in surface loss per year, but the relative surface loss has increased. Oppositely, the rates of glacier thickness loss have increased during the study period ($-0.6\,\mathrm{m\,yr^{-1}}$ from 1981 to 2011 and $-1.1\,\mathrm{m\,yr^{-1}}$ from 2011 to 2022), indicating an acceleration of glacier ice thickness loss, especially in the last decade, and more pronounced in the last 3 years. In terms of specific mass balance (considering only changes at the smallest surface glacier, the most recent year

of comparison), the losses are $0.6\,\mathrm{m\,w.e.\,yr^{-1}}$ for the period 1981–2022, $0.5\,\mathrm{m\,w.e.\,yr^{-1}}$ for the period 1981–2011, $1\,\mathrm{m\,w.e.\,yr^{-1}}$ for the period 2011–2022, $1.2\,\mathrm{m\,w.e.\,yr^{-1}}$ for the period 2020–2021 and $2.7\,\mathrm{m\,w.e.\,yr^{-1}}$ for the period 2021–2022. Based on these results, two inflexion points can be identified, one after 2011 and another after 2020 – in both cases the thickness loss has accelerated sharply. This ice thickness loss is mainly accelerated (among other factors) by the fact that the accumulation zone over the glacier in summer is negligible, especially during very hot summers as in 2022, and the ablation zone covers the entire glacier, as no ELA is observed for some years. Unfortunately, due to the small extent of this glacier no reliable satellite observations of sufficient resolution are available for the Aneto glacier ELA in late summer, and this absence of accumulation area is based on field work observations of UAV operators.

Various studies of other glaciers in the Pyrenees have also shown a continuous increase in glacier thickness and area

losses, with a high interannual variability but a clear negative trend over longer time periods. These works focused on the Monte Perdido glacier (López-Moreno et al., 2019), Ossoue glacier (Gascoin and René, 2018), Maladeta glacier (Pastor Argüello, 2013) and La Paul glacier (Rico et al., 2015). Hugonnet et al. (2021) also determined a mean ice thinning of $-0.96\,\mathrm{m\,yr^{-1}}$ for the Pyrenean glaciers for the period 2000–2019. Although this work focuses on the period 1981–2022, the glaciers of the Aneto–Maladeta massif had about 610 ha at the end of the LIA, so they lost about 338 ha from 1850 to 1984 (Rico et al., 2017).

The mean annual specific mass balance values of $-0.6\,\mathrm{m\,w.e.\,yr^{-1}}$ on the Aneto glacier determined for the period 1981–2022 are similar to those in other studies in the Alps, such as Davaze et al. (2020), who estimated an annual mass balance of $-0.7\,\mathrm{m\,w.e.\,yr^{-1}}$ from 2000 to 2016 for 239 Alpine glaciers. Similarly, Carturan et al. (2016) determined the mean annual mass balance of nine Italian glaciers from 2004 to 2013, which ranged from $-1.8$ to $-0.8\,\mathrm{m\,w.e.\,yr^{-1}}$. This is also supported by other climatic data showing an increase in air temperatures over the past century (Bolch et al., 2012; Rabatel et al., 2013), particularly a sharp rise in temperatures at high elevations and low latitudes (Vuille et al., 2008; Pepin et al., 2015), accompanied by a shorter duration of seasonal snow cover (Brown and Mote, 2009).

Vidaller et al. (2021) describe the changes in ice thickness of the Aneto glacier (among other glaciers of the Pyrenees) based on an ice thickness decrease of 8.5 m during the period 2011–2020. Based on the ice thickness reconstruction data of the Aneto glacier presented in this study, the mean ice thickness in 2020 was 15.0 m, while in 2011 it was 19.2 m, so the loss is 4.2 m. This difference is due to the fact that the mean ice thickness of 2011 was calculated based on the extent of 2011, and the ice thickness of 2020 was calculated based on the area of 2020, while in the case of Vidaller et al. (2021) the ice thickness loss was calculated considering only the ice thickness loss within the glacier area of 2020. A similar problem exists when comparing the remaining ice thickness in 1981 (32.9 m) and in 2022 (11.9 m) with the ice thickness losses for the period 1981–2022 (30.5 m). The remaining ice thickness in 1981 is similar to those losses calculated for the period 1981–2022; meanwhile the remaining average ice thickness was 11.9 m. The mean ice thickness for a particular year was calculated based on the extent observed for that year.

## 4.2 The importance of the methods

Remote sensing techniques have developed rapidly in recent years, allowing observation of the Earth's surface with a spatial resolution that was previously impossible. This work exploits historical aerial photographs to reconstruct a digital surface model for the year 1981 and provides a comparison to observe changes in landscapes and surfaces in detail.

Campos et al. (2021) calculated changes in the Aneto glacier from the LIA to 2017 using data from 1957, 1983, 2000, 2006, 2015 and 2017. In 1983, they reported an area of 103.2 ha (1.03 km$^2$), in contrast to the 135.7 ha (1.36 km$^2$) for 1981 described in this work. The large difference may be due in part to the fact that they did not consider the slope angle of the terrain in their calculations (2D vs. 3D surface). Nonetheless, considering our delineation, but ignoring the effect of slope angle on the area estimate, we would have reported a value of 115.5 ha (1.16 km$^2$) for 1981, which underestimates our value by 20 %. This study also uses the National Fly photograms to convert to point clouds, accounting for stable GCP during the study period. This is a more accurate method because it avoids distortion of the Plan Nacional de Ortofotografía Aérea (PNOA) orthophotos used by Campos et al. (2021), who acknowledge a source of uncertainty: "The extension for the 1983 stage should be considered with caution, due to the lower quality of the 1983 aerial Image" CE10 . The area determined in our study is closer to that reported by Arenillas-Parra et al. (2008), who reported an extent of 136 ha (1.36 km$^2$) for the Aneto glacier in 1982 based on aerial photographs of a specific flight in the glaciated areas of the Pyrenees.

The values of ice thickness from the GPR reported in Campos et al. (2021) also show significant differences not consistent with our results. In 1994, the ERHIN CE11 programme estimated a maximum ice thickness of 52 m using 17 transects spaced 100 m apart (Arenillas-Parra et al., 2008; Jiménez-Vaquero, 2016). In 2008, those authors determined a maximum ice thickness of 30 m calculated from 31 GPR transects (Jiménez-Vaquero, 2016). Considering these data, Campos et al. (2021) reconstructed the subglacial topography of the Aneto glacier, and based on this topography they determined a maximum ice thickness of 55 m for 1983, 37 m for 2006 and 29 m for 2015. These values are in stark contrast to our estimates (maximum of 96.5 m in 1981, 52.5 m in 2011, 44.7 m in 2020, 43.5 m in 2021 and 41.8 m in 2022). Comparing the values of remaining thickness reported in 2008 (maximum ice thickness of 30 m; Jiménez-Vaquero, 2016) and the rate of ice thickness loss ($-1.0\,\mathrm{m\,yr^{-1}}$) established by Vidaller et al. (2021) for the period 2011–2020, the expected maximum thickness in 2020 would be 18 m instead of the 44.7 m we observed in 2020. Additionally, large areas currently covered by the glacier would be ice free according to the previous ice thickness loss estimates. Considering that we used comparable values for wave propagation velocity of GPR signal to those used in the above-cited work, the differences between previous literature studies and the glacier thicknesses reported here are likely related to the more modern and accurate antennas used in our survey and the much denser net of transects conducted in the 2020 campaign. This methodology significantly reduces the uncertainties associated with the interpolation process, making the results obtained here more robust, and also permits a better understanding of the glacier's dynamic and its future behaviour.

### 4.3 Future perspectives

The rate of surface and ice thickness losses calculated in this study and the reconstruction of ice thickness for the year 2022 indicate the critical situation of this glacier. There are no signs of slowdown in glacier surface and thickness loss rates; on the contrary, we have observed the high vulnerability of the Aneto glacier to the occurrence of extremely hot summers in recent years, as in 2022, when summer temperatures were $0.5\,°C$ above the mean for the period 2007–2022, according to the Renclusa station (2140 m a.s.l.), and almost $2\,°C$ in general in the Iberian Peninsula (AEMET). Thus, the continued loss of surface area and thickness could be due to an increase in temperature.

Taking into account the average current glacier thickness of 11.9 m, we can affirm that the Aneto glacier is indeed in its terminal stage, with evident fragmentation into smaller ice bodies, the absence of a significant accumulation zone and obvious signs of ice stagnation. In this context, glacier retreatment is exposing new areas of unconsolidated bedrock material (granite boulders and debris) and destabilising cirque walls in many areas. This process is also accompanied by a degradation of surrounding wall permafrost (Rico et al., 2021). Under this situation the occurrence of unusual warm periods CE12, such as those observed in the 2021–2022 period, triggered hazardous rockfalls, as were also noticed in other mountain areas (Huggel et al., 2010; Kellerer-Pirklbauer et al., 2012). This behaviour could also anticipate the behaviour of other temperate mountain glaciers in their final deglaciation phases.

Another aspect that determines the evolution of the Aneto glacier is the darkening of the glacier surface. However, a detailed quantification of the darkening of the glacier surface and its effect on the energy and mass balance has not been carried out yet. Early spring (summer) snowmelt and glacier thickness loss result in a grey (dark) appearance of the glacier surface, which reduces the albedo effect and increases the absorption of thermal energy, leading to an acceleration of glacier surface and thickness losses (Shaw et al., 2021). The obvious similarities with the remaining glaciers of the range suggest that the Pyrenees may become an ice-free mountain range in the next few decades.

The rise in temperature in recent decades, combined with a slight decrease in precipitation, has resulted in less snow accumulation during the winter months. This results in longer exposure of the glacier during the ablation season, which increases the melting of the glacier from year to year. Compared to Pyrenean glaciers that have a minimal contribution to water resources in downstream areas (López-Moreno et al., 2020; Milner et al., 2017), changes in snowpack can lead to severe changes in the downstream water regime (García-Ruiz et al., 2011).

Also of note is the presence and development of new proglacial lakes, as in the case of Ibón Innominato. This small lake is in constant change due to the surface and thickness of the glacier, where the retreat of the glacier front has opened new outlets beneath the glacier, and consequently the water level of the lake decreases. Similarly, as the Aneto glacier shrinks, other lakes would be formed in the depression areas derived from the subglacial topography. The presence of proglacial lakes negatively affects the glacier's equilibrium by acting as an energy collector and accelerating the rate of thawing at the front of the glacier (Otto, 2019). In addition, the dark appearance of the glacier surface caused a decrease in albedo and therefore an increase in the surface and thickness losses of the glacier (Yue et al., 2020).

On the other hand, the maximum ice thickness ($> 44$ m) is located under the Maldito pass, a protected area fed by avalanche channels and protected by the shadow of the Maldito peak. In these areas, longer persistence of the ice body is expected.

The fast surface loss of the Aneto glacier in the last few decades and the relatively low ice thickness observed together with the potential development of new lakes clearly show the consequences of climate change in mountain areas. Those changes happening nowadays in most mountain glaciers (Kääb et al., 2021; Barrand et al., 2017; DeBeer and Sharp, 2009) will have a major impact on mountain landscapes and ecosystems (Huss et al., 2017), showing the necessity of monitoring and understanding the recent fast evolution of these environments.

How long the glacier will maintain the main the ice movement CE13 and a surface greater than 2 ha to still be considered a glacier is a very uncertain issue to be estimated. The duration of the glacier depends on several factors, such as the temperature evolution in the next few years, the evolution of precipitation (mainly snowfall in winter), the ability of the glacier to transport the debris fallen from the headwalls (and avoid the darkening of the surface), possible events of dust deposition (which may be frequent in winter and spring) and many other factors. In addition, according to the study by Vidaller et al. (2021), it is possible that these very small glaciers, once they become smaller than 10 ha, will have a greater topoclimatic control, so their preservation could be prolonged if there are no more very hot summers, as in 2022. Otherwise, glacier extinction could be imminent if there are a few summers like 2022 in the next decade. However, more detailed studies are needed to answer such a simple question to reduce the uncertainty in observations and simulations and also to provide a deeper understanding of those processes that govern small and very small glaciers.

### 5 Conclusions

The Aneto glacier, although it is considered a very small glacier, is the largest glacier in the Pyrenees and also the largest in southern Europe. However, climate change has accelerated its disappearance, in line with other glaciers in the range. The evolution of close-range remote sensing tech-

niques allowed us to observe the glacier surface in a very high level of detail that permits comparison between different years' surface (DEMs) of the glacier and evaluation of its changes.

For the period 1981–2022 the Aneto glacier surface has diminished 64.7 % (from 135.7 ha (1.36 km$^2$) to 48.1 ha (0.48 km$^2$)), and its front has shifted from 2828 to 3026 m. It has also been divided into two bodies between 2015 and 2016, and a proglacial lake has appeared in front of it in the last few years. The annual rate of surface loss has been constant over time ($-2.2$ ha yr$^{-1}$), but the relative surface loss of the glacier surface has increased during the study period.

The mean ice thickness loss was estimated at 30.5 m for this 41-year period (with maximum losses over 80 m), with a specific mass balance of $-0.7$ m w.e. yr$^{-1}$. However, the annual specific mass balance ratio has been increasing; in fact it quadrupled ($-2.7$ m w.e. yr$^{-1}$ for the period 2021–2022) over the period 1981–2022. Using GPR measurements, we have estimated a mean of 44.7 m of ice thickness in 2020. GPR data and ice thickness loss estimated with UAV data have been used to infer the actual mean ice thickness, which was 11.9 m. The ice thickness distribution shows areas around the glaciers with very little thickness ($< 2$ m), so these zones are very close to becoming deglaciated during the coming summers. The surface and thickness losses of the Aneto glacier indicate the critical situation of this ice mass. It is in its terminal stage, displaying fragmentation into smaller ice bodies and the presence of debris cover in some areas.

*Data availability.* At the time of publication, the database of glacier thickness changes and glacier delimitation in 1981, 2020, 2021 and 2022 will be available through https://doi.org/10.5281/zenodo.7472185 (Vidaller et al., 2022).

*Supplement.* The supplement related to this article is available online at: https://doi.org/10.5194/tc-17-1-2023-supplement. TS9

*Author contributions.* Conceptualisation: IV, JILM, EI, JR; methodology: IV, EI, LMdR, JR; software: IV, EI, JR; validation: IV, EI, LMdR, JR, JILM; formal analysis: IV, JR, EI; investigation: all authors; resources: JILM; data acquisition: all authors; writing (original draft preparation): IV; writing (review and editing): all authors; visualisation: IV; funding acquisition: JILM. All authors have read and agreed to the published version of the paper.

*Acknowledgements.* This work was supported by the Interreg-POCTEFA project OPCC ADAPYR, the Spanish Ministry of Economy and Competitiveness project CGL2017-82216-R, and the Spanish Ministry of Science and Innovation PID2020-113247RB-C21 and PID2021-124220ob-100/MARGISNOW. Jesús Revuelto has been supported by the IJC2018-036260-I (Juan de la Cierva I) and RYC2021-033859-I (Ramón y Cajal) projects. Ixeia Vidaller is supported by the grant FPU18/04978 and is enrolled in the PhD programme at the University of Zaragoza. Eñaut Izagirre is supported by the grant PPGI19/02 (UPV/EHU) and the Consolidated Research Group IT1678-22 (Basque Country Government). Esteban Alonso-González has been funded by the CNES postdoctoral fellowship. TS10We thank the Spanish National Geographic Institute (IGN) for the collection, archiving and distribution of the aerial photographs. We also thank the NextGIS/QuickMapServices plugin (original work published in 2014), available online at https://github.com/nextgis/quickmapservices (last access: 2 June 2021), and AEMET for sharing the climatic data of the Renclusa hut.

*Financial support.* This research has been supported by the NAME OF FUNDER (grant no. GRANT AGREEMENT NO). TS11

We acknowledge support of the publication fee by the CSIC Open Access Publication Support Initiative through its Unit of Information Resources for Research (URICI).

*Review statement.* This paper was edited by Nicholas Barrand and reviewed by Pierre Pitte and one anonymous referee.

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

**Remarks from the language copy-editor**

CE1    Please note that Zaragosa was changed to Saragossa as per a house standard.

CE2    As per a house standard, when generic terms are used descriptively, or used alone, they are lower-cased (e.g. the Pacific coast, the Californian desert; Chicago p. 408).

CE3    This term has been changed according to Copernicus's policy on gender-neutral language and political correctness.

CE4    Please verify all the slashes used throughout this paper. A slash can be used to mean "or", but if you meant "and" or "and/or" in some or all instances, the slash should be replaced. Please check all cases and indicate whether any instances should be changed.

CE5    Please check that the meaning of your sentence is intact.

CE6    Please check that the meaning of your sentence is intact.

CE7    This abbreviation is not defined. Is it well known or should it be defined for clarity?

CE8    Please check this sentence for completeness.

CE9    Do you mean "depression of" or would that change your intended meaning?

CE10    Is "Image" capitalised in the original quote? If not, it should be lowercased.

CE11    This abbreviation is used only once. Is it well known or should it be defined for clarity?

CE12    Do you mean "unusually warm periods" or would that change your meaning?

CE13    Please rephrase "the main the ice movement" for clarity.

**Remarks from the typesetter**

TS1    The composition of Figs. 3–5 has been adjusted to our standards.

TS2    Please provide date of last access.

TS3    Please note that units have been changed to exponential format throughout the text. Please check all instances.

TS4    Fischer (2009) is not in the reference list.

TS5    RGI Consortium (2017) is not in the reference list.

TS6    Jimenez (2016) is not in the reference list. Do you mean Jimenez-Vaquero (2016)?

TS7    López-Moreno et al. (2018) is not in the reference list.

TS8    Palacios et al. (2020) is not in the reference list. Do you mean Palacios et al. (2022)?

TS9    Please send a new Supplement as a *.pdf without the title, authors, correspondence author, etc. as we will generate a Supplement title page during publication (with a citation including the DOI), which will contain this information.

TS10    Please note that the following part has been moved here because it fits better into the acknowledgements.

TS11    Please note that there is funding information given in the acknowledgements, but you did not indicate any funding upon manuscript registration. Therefore, we were not able to complete the financial support statement. Please provide the missing information and double-check your acknowledgements to see whether repeated information can be removed from the acknowledgements. Thanks.

TS12    Please ensure that any data sets and software codes used in this work are properly cited in the text and included in this reference list. Thereby, please keep our reference style in mind, including creators, titles, publisher/repository, persistent identifier, and publication year. Regarding the publisher/repository, please add "[data set]" or "[code]" to the entry (e.g. Zenodo [code]).

TS13    Please provide the complete author list.

TS14    Please provide the page range or article number.

TS15    Please provide the journal name.

TS16    Please provide the page range or article number.

TS17    Please provide the page range or article number.

TS18    Please provide the editors (if not authors) and a persistent identifier.

TS19    Please provide the editors (if not authors), the publisher and a persistent identifier.

TS20    Please provide the volume and page range/article number.

TS21    Please provide the page range or article number.

TS22    Please provide date of last access.

TS23    Please provide the complete author list.

TS24    Please provide the page range or article number.

TS25    Please provide the page range or article number.

TS26    Huss and Hock (2018) is not mentioned in the text.

TS27    Please provide the complete author list.
TS28    Please provide the editors (if not authors) and a persistent identifier.
TS29    Please provide the editors (if not authors) and a persistent identifier.
TS30    Please provide the complete author list.
TS31    Please provide the volume and page range/article number.
TS32    Please provide the complete author list.
TS33    Please provide the complete author list.
TS34    Please provide the page range or article number.
TS35    Please provide the editors (if not authors) and a persistent identifier.
TS36    Palacios et al. (2022) is not mentioned in the text.
TS37    Please provide the complete author list.
TS38    Please provide the page range or article number.
TS39    Please provide the complete author list.
TS40    Please provide the page range or article number.
TS41    Please provide the page range or article number.
TS42    Please confirm.
TS43    Please provide the date of the conference and the poster number.
TS44    Please provide the journal, volume and page range/article number.

**Please note the remarks at the end of the manuscript.**

**The Cryosphere, 17, 1–16, 2023**                                                  **https://doi.org/10.5194/tc-17-1-2023**