# Peer review of "The Aneto Glacier (Central Pyrenees) evolution from 1981 to 2022: ice loss observed from historic aerial image photogrammetry and remote sensing techniques"

_The Cryosphere, 2022_

## Referee Comment (RC2)

**Review of 'The Aneto Glacier (Central Pyrenees) evolution from 1981 to 2022: ice loss observed from historic aerial image photogrammetry and recent remote sensing techniques' submitted by Vidaller et al for publication in *The Cryosphere***

In this manuscript, the authors examine the historic and recent geometric changes to Aneto Glacier, the largest remaining glacier in the Central Pyrenees. They leverage a variety of topographic data sources to measure area and thickness changes since 1981, with a focus on the last 3 years due to the availability of UAV surveys. They also conducted field GPR measurements of ice thickness to map the contemporary bedrock and reconstruct the current and historic ice thickness. They then use the former to assess bedrock topographic variability and possible overdeepenings.

In general the assemblage of field measurements and historic surveys provides a rich overview of changes to Aneto Glacier, and the documentation of the demise of an iconic Pyrenean glacier will definitely be a worthwhile contribution to the literature. However, I have some major concerns relating to several aspects of the manuscript, all of which should be addressed before the manuscript will be suitable for publication.

1. Research aim. The research aims need to be reformulated somewhat. I also do not think the authors have addressed their aims in the discussion: e.g. anticipating the evolution of other temperate mountain glaciers (L70) or determining the inflexion point in glacier mass loss (L85). Meanwhile 'showing' the speed of glacier change is not really a scientific objective.

2. Clarity of presentation of results. The authors refer to thickness change, but in some cases their meaning is not entirely clear or consistent with the literature. I believe that they are determining the slope-perpendicular surface change (as normally derived from the M3C2 algorithm in CloudCompare), but this differs strongly from the typical surface height change values reported from geodetic studies (e.g. Hugonnet et al., 2021) and the authors need to be clear about that difference, as well as to check that it is what they wish to present. I would recommend also presenting traditional thinning values as well as the total volumetric change of the glacier for each period of interest. This feeds into the need to improve the discussion/justification for considering surface areas vs planimetric areas (again, I recommend reporting both values), which are useful for distinct analyses. More importantly, in some cases the authors seem to equate mean changes in ice thickness change (ie thinning) with changes to the mean ice thickness (e.g. L265-269, L389-392); these quantities differ fundamentally due to changes in glacier area due to thinning, and the authors need to reconsider these comparisons.

3. Gaps in presentation of the results. The authors have compiled visually aesthetic figures, in general, but I think there are some fundamental gaps in their presentation of results. I would specifically request that they present the GPR transects in the main manuscript, along with an assessment of the overall uncertainty of their bedrock estimates (partial uncertainty from the crossing points in Table S2, but also consider the radar uncertainty in general), provide thinning maps for each period of the UAV surveys, include off glacier measured elevation changes (Figure 3 appears cropped) and an empirical uncertainty assessment of the thinning rates, which is necessary as the authors do not seem to have used independent GCPs to assess the accuracy of their UAV-derived elevation models. I also think that the thinning/volume change results would benefit from a temporal summary plot (also depicting uncertainties), as in e.g. (Ragettli, Bolch, & Pellicciotti, 2016).

4. Demonstration of the suitability of TPI for identifying bedrock overdeepenings. This is an interesting idea as an alternative to topographic-routing approaches to identify sinks, but the authors need to demonstrate that it is useful for this purpose – does this approach work

to identify bedrock sinks in mountainous terrain? Analytically-speaking, I am concerned that, as TPI is also dependent on local topographic high values, the sinks that the authors identify are actually due to the neighbouring high topography of glacier headwalls, rather than locally low values of bedrock topography. The authors would also do well to compare to a conventional drainage-based identification of topographic lows.

5. Discussion. The discussion could be improved in several ways, and ideally should provide new insights. At present, the comparison to past studies (both Vidaller et al, 2021 and Campos et al, 2021) does not provide a clear, consistent explanation for discrepancies between values reported in past studies (note also my point 2 above relating to potential discrepancies between the physical variables measured, as well as the lacking rationale for surface area instead of planimetric area). This should all be clarified, but more importantly the Discussion should be geared to critically evaluate their own results and to provide new insights. The authors make a very casual assertion of albedo change and melt season duration prolongation, but without evidence, and attribute the accelerated mass losses to warming, without presenting analysis of a climatic record. I also miss a critical evaluation of the possibility of lakes based on the TPI results. Most importantly, though, I feel the study misses a chance to postulate on the future changes of this system. As select examples: at the current rates of volume loss, how long do we expect Aneto to last? When would the hypothesized overdeepenings become exposed? Is the headwall likely to provide a microclimate that can prolong the glacier's life? Or, indeed, does Aneto still qualify as a glacier – is it moving dynamically? – and when may it be too small to be considered as such? Although the field data they have collected is commendable, I feel that the authors need to go further examining some of these aspects if they wish this study to be a meaningful contribution to the literature.

Minor comments:

L21. Please check 'glaciated' vs 'glacierized'

L39. (Huss & Hock, 2018) is a great modelling study but is not 'observed'

L44. Misplaced colon :

L48. I suggest refraining for using 'ice thickness wastage', which is a vague term, and instead using volume loss or thinning, here and throughout the manuscript. Similarly, for clarity I would recommend 'area loss' instead of shrinkage.

L49. Can you be more specific about which volume change data is lacking? Clearly there is now the (Hugonnet et al., 2021) dataset, but perhaps this is not sufficiently precise or long-term?

L52-53. Please refer to thinning rates for this comparison.

L55. 'considered' is redundant here

L59. 'having an additional protection figure' – unclear what you mean here – I suppose that the authors refer to additional societal value?

L66. 'comprehension' -> analysis

L67. Is this the largest spatio-temporal dataset of glacier observations? I don't think that is necessary motivation for this study

L68-71. Please consider the scientific objectives: 'showing' consequences is not a valid scientific motivation (implies bias), but investigating/measuring/understanding is.

L72. Please use either 'subject of annual monitoring' or 'subjected to annual monitoring'

L91. I do not understand what is meant by 'sectors can be delineated…'

L94. 'longest period ever observed' seems like hyperbole – have there been investigations of the LIA extent? LGM?

L108. Is this the annual isotherm or a seasonal isotherm?

L110. This formulation of the temperatures is not very clear to me (L112-115 for the Aneto summit is much clearer). Over what period are these the maximum and minimum mean annual temperatures?

L141 Please check how to acknowledge this source in the 'Acknowledgements' section of the manuscript. Can you ascertain any information with regards to the precision of geospatial positioning of the data?

L142. Do you use any independent check points?

L159. How does using the same protocol imply reduced uncertainties? I can imagine the argument that it implies similar uncertainties to a prior study, but this should be borne out with direct measurements.

L162-3. Please rephrase this statement – have you then assessed the distribution of deviations on stable ground? Is this from individual check points or the distribution of absolute deviations, or?

L192. Have these data been submitted to GlaThiDa (Welty et al., 2020)?

L199. (also 237-239)How was the division done? Is this totally random, or patch-wise? Note that your radar returns are nearly continuous and autocorrelated, so a random subsetting is likely to give a very good cross-validation result, but has little to no information with regards to the accuracy of your interpolated product.

L207. Strong slope = steep. Please give a slope value?

L211. Is this the 3D or xy geolocation accuracy? I disagree that using the same protocol will give the same results – this is dependent on weather conditions (especially wind and humidity) as well as the stability of camera focus (for example).

L217. M3C2 gives the surface-perpendicular distance, rather than thinning. Does it also give a total-volume-change estimate?

L222. Please note the uncertainty of this assumed value.

L225. The TPI method is an interesting idea, but it is not very clearly justified – in particular because TPI values are extremely low directly beneath steep areas, whether or not the location is a topographic low. Is there a justification for using TPI instead of a drainage model, as is commonly used (Linsbauer, Paul, & Haeberli, 2012)?

L240-242. Do you have an estimate of the uncertainty of delineation from the 1981 survey?

L245. 300% larger than?

L251. By 'no movement' you seem to mean that the terminus is not retreating higher, is that correct? Or did you assess surface velocity?

Table 1. Please consider providing uncertainty estimates for your area-change assessments.

L265. Does this 'mean ice thickness loss' correspond to the thinning (i.e volume change over the glacier area) or the change in the mean ice thickness over time? These are two very different properties that need to be clearly disambiguated (both are worth noting!).

L275. Please consider presenting volume changes (loss is a negative number).

Figure 3. If I understand correctly, this figure presents the surface lowering (ice thinning). Please also indicate the extents of the glacier in 1981 and 2022 on the upper panel.

Figure 4. Please indicate the locations of the GPR lines on this left panel. On the right panel, I would consider a box plot rather than the range of thicknesses (the dots), which will give a better sense of the distribution of thicknesses for each elevation

L298-300. This is Methods.

L302-303. Glacier thickness was high -> glacier was thick

L306. Please use past tense also for the 2022 observations.

Figure 5. Panel A looks quite noisy due to localized artefacts in the 1981 dataset, which you could probably justify filtering out. Could you use these reconstructed geometries to look at the volume-area relationship for this glacier (Bahr, Pfeffer, & Kaser, 2015)?

L325. Misplaced period after the Westoby citation

L325-330. I think some more justification and demonstration for the TPI method is needed – can you validate the approach with now-exposed overdeepenings, for instance the Indominato Lake from the 1981 data? How does this compare to a flood-filled glacier bed DEM? How is this method better for identifying overdeepenings?

Figure 6. The locus of low TPI values below the headwall is due to the headwall, rather than due to the subglacial topography.

L344. Was there any accumulation area in 2021 or 2022?

L345. Neither PlanetScope nor Sentinel-2 are sufficiently high resolution?

L358-367. I feel that some of this discussion could simply be confusion about what a previous study reported, and how this calculation works. The mass balance is the mean value of thinning, not the change of the mean thickness. It is clear that the mean thickness will not decrease at the same rate as the surface thins, because the mean thickness is only calculated for the glacier area that is still glacier. Please reconsider this section. I would suggest to also refer to the Hugonnet et al (2021) results for this domain, even if they are imprecise due to resolution and so on. In fact, I would strongly recommend a figure compiling and comparing the mass balance observations available from this study, Vidaller et al (2021), Campoes et al (2021) and Hugonnet et al (2021), as well as the observed spatial extents.

L374-382. I have a hard time understanding the authors' argument here – what is the advantage that they see in investigating surface area changes instead of planimetric changes? As there is a considerable deviation due to the high surface slopes, should the authors not simply report both?

L388. Have the authors compared the spatial distribution of measured ice thicknesses from their own observations and Campos et al (2021)? I see clearly that there is a statistical difference but the authors need to demonstrate that this is not simply due to spatial biases in sampling. How do the observed bedrock elevations compare in space?

L401. This link to hot summers in recent years has not been explicitly made, and would require a comparison of (best if long-term) measured mass balance along with climate information. Although this is very likely to be true, the authors have not tested or demonstrated this, only that the glacier is continuing to rapidly lose mass.

L409-414. Please write this more clearly, e.g. heat -> energy. Also, have you evaluated the albedo changes (not demonstrated here), or is this a hypothesis? Same for the longer exposure of the glacier – please clearly differentiate between observations and conceptual discussions.

L411-412, L424-426 The references to Shaw et al (), Otto (), and Yue et al () need to be reformatted.

L450. Realizing that this would entail a change of scope, I think it would be very worthwhile to also consider the future evolution of this glacier and its neighbours. See e.g (Huss & Fischer, 2016).

L478. Alignment issue with this reference

L470 – Several reference formatting issues. Please ensure that you follow the guide for The Cryosphere.

References:

Bahr, D. B., Pfeffer, W. T., & Kaser, G. (2015). A review of volume-area scaling of glaciers. *Reviews of Geophysics*, *53*(1), 95–140. https://doi.org/10.1002/2014RG000470

Hugonnet, R., McNabb, R., Berthier, E., Menounos, B., Nuth, C., Girod, L., … Kääb, A. (2021). Accelerated global glacier mass loss in the early twenty-first century. *Nature*, *592*(July 2020). https://doi.org/10.1038/s41586-021-03436-z

Huss, M., & Fischer, M. (2016). Sensitivity of very small glaciers in the swiss alps to future climate change. *Frontiers in Earth Science*, *4*(April), 1–17. https://doi.org/10.3389/feart.2016.00034

Huss, M., & Hock, R. (2018). Global-scale hydrological response to future glacier mass loss. *Nature Climate Change*, *8*(2), 135–140. https://doi.org/10.1038/s41558-017-0049-x

Linsbauer, A., Paul, F., & Haeberli, W. (2012). Modeling glacier thickness distribution and bed topography over entire mountain ranges with glabtop: Application of a fast and robust approach. *Journal of Geophysical Research: Earth Surface*, *117*(3), 1–17. https://doi.org/10.1029/2011JF002313

Ragettli, S., Bolch, T., & Pellicciotti, F. (2016). Heterogeneous glacier thinning patterns over the last 40 years in Langtang Himal. *The Cryosphere*, *10*(5), 2075–2097. https://doi.org/10.5194/tc-2016-25

Welty, E., Zemp, M., Navarro, F., Huss, M., Fürst, J. J., Gärtner-Roer, I., … Li, H. (2020). Worldwide version-controlled database of glacier thickness observations. *Earth System Science Data*, *12*(4), 3039–3055. https://doi.org/10.5194/essd-12-3039-2020

---

## Author Comment (AC1)

Dear Editor, dear Reviewers:

We are pleased to submit a revised version of our manuscript entitled "The Aneto Glacier (Central Pyrenees) evolution from 1981 to 2022: ice loss observed from historic aerial image photogrammetry and remote sensing techniques" by Vidaller et al where we have incorporated your suggestions and corrections. We thank you very much for the valuable comments that certainly helped to improve our manuscript. Please, note that we have also simplified the title by removing the term "recent" as this was not informative nor accurate enough.

Below we provide a point-by-point response to all comments raised in the reviews. We hope that the revised manuscript now meets the quality standards of *The Cryosphere*. I will be happy to answer any question you might have regarding this study.

We look forward to hearing from you.

Sincerely,

Ixeia Vidaller and co-authors

General comments

I find this paper to be a convincing contribution to the current state and recent evolution of the Aneto glacier in the Pyrenees. This is a well-studied glacier, with many previous studies, but the current paper puts together an impressive and updated series of datasets. The work is nicely and extensively illustrated as well.

The methods are consistently explained and documented.

On the hind side, I think the results read with some difficulties due to the emphasis in including lots of data for each statement. The text could be made simpler by making better use and focusing on the trends shown by the figures. If needed, the detailed data of area, thickness and depth changes can be included in tables in the supplementary material.

From a methodological point of view, this work integrates data from gridded data sets (DEM), point clouds (UAV DEM) and transects (GPR) with variable spatial footprints. Maybe consider a resampling of the different datasets to a common gridded base, which would make the integration easier, both in terms of the representation of the results in the figures and maps, and in the explanations in the results and discussion sections.

I consider the paper would benefit from minor correction prior to publication.

Answer (hereafter A):

We thank RC 1 for a very constructive and positive review of the manuscript. Indeed, there are lots of data for each stamen, so to avoid repeat these data in the text and in table, we have move surface and thickness changes to the *Supplementary material*.

Also, we certainly agree on the limitations resulting from the different spatial resolution of the methods used in this study. Thus, and with the aim of avoiding possible misunderstandings, we have modified some figures to facilitate their interpretation. Please review Figures 2, 3 and 5 on this regard. Following the RC1 recommendation about the text in figure captions, we have better described the results of the hypsography changes in Figure 3 caption:

> *"Figure 3: (A) Thickness loss of Aneto Glacier from 1981 to 2022. In the upper map, the black line delineates the glacier in 2022 while grey line represents the glacier in 1981. The arrow indicates the north direction. (B) Distribution of thickness loss considering elevation bands of 20 m."*

**In-line comments**

Line 47. Here and elsewhere glacier areas should be expressed in km$^2$ (WGMS, 2009). While it is not incorrect to use ha, km$^2$ communicates the relative size of the studied glacier

A: Thank you for the appreciation. Due to the small size of all Pyrenean glaciers, it is common to use ha as the unit of measure, as in Rico et al. (2017) and Vidaller et al. (2021), for example. However, in light of your comment, we have included the area in km$^2$ in parentheses, to make it easily comparable to other previous published data.

Figure 1. I suggest including political boundaries in A) and keep the same line type in B). In B) increase the hillshadding so that the relief is highlighted. Also, I would expect glacier outlines rather than point indicating glacier position here (handle line thickness to make glacier visible at this scale). I think C) is a nice and very clear way to indicate the main geographic features of the study area.

A: We discard adding political boundaries since they could overload the map of Europe too much and compromise the clarity. Therefore, we prefer not to include them in A). Nevertheless, Figure 1B) includes the Spanish-French border.  However, if RC1 still thinks that borders are needed in A), we are open to focus this map on a smaller area. Since the glaciers of the Pyrenees are too small (all <50 ha), we decided to assign points to each site instead of marking their surfaces to better identify their location in a 50 x 200 km map.

[Figure]

*"Figure 1: Location of the Aneto Glacier. (A) Map of Europe, with the pink rectangle delimiting the central part of the Pyrenees. © Google Maps (B) Topographic map of the central Pyrenees; the glaciers in this area are marked with grey dots and the location of the Aneto Glacier is marked with a pink star. (C) An aerial photo of Aneto Glacier in summer 2021. The main reliefs surrounding the glacier are indicated."*

Line 108. I suppose this is the mean annual isotherm, please clarify. Consider including a climograph with the station data. This kind of graph clearly shows the magnitude and seasonality of the main climatic variables. Finally plot the stations location in Figure 1 B).

A: Amended as suggested. The Renclusa station is not visible in Figure 1C and, due to the extent of map in Figure 1B, the location of this station overlaps with the location of the glacier. In such a situation, we cannot plot the Renclusa station location.

On the other hand, we have finally decided not to include climate diagrams as an additional figure since the number of figures in the manuscript is already too large. However, in response to a comment made by RC2, we have enlarged the text with climatic information to fill the indicated gap. Still, the altitude difference between the Renclusa station and the glacier and the short period when observations are available in the Aneto station (2018-2022 years), precluded an in-depth analysis of these climatologies. Therefore, we focus our attention on the evolution of the glacier. This is show in the main manuscript as (L127-130):

*"Mean annual temperature for the period 2007-2022 was 4.6 ºC at the weather station of Renclusa hut (2,140 m a.s.l.), meanwhile the mean temperature for the same period in the ablation season (June-September) was 11.6 ºC. 2022 was an especially warm year, in which annual mean temperature was 5.2 ºC and the summer mean temperature was 12.1 ºC (data from the AEMET database)."*

Line 162. Do you mean mountain areas? Or heterogenous in terms of ground cover? Please clarify.

A: Thank you for the appreciation. We mean mountain areas, in terms that is an irregular terrain with changes in elevation and slopes.

For clarification, we have reworded the text as follows:

*"shows equivalent accuracies for working in highly heterogeneous areas"* → *"shows equivalent accuracies to work in this area".*

Line 204-208. While a glacier "true" area is better approximated by a 3D approach, most glaciers inventories report projected areas and this is the standard approach (WGMS, 2009) so, by making this choice you make your data harder to integrate with most other regional and global datasets.

A: We agree with this comment and understand the difficulty to integrate our data with global datasets but, at the same time, the small size of Pyrenean glaciers forced us to use the 3D area, thus gaining a more accurate representation of their real surface. To facilitate comparison with other datasets and following this reviewer suggestion, we have included in Table 1 (now in the *Supplementary Material Table S5*) another column with the 2D area data.

**Table S5:** Main characteristics of the Aneto Glacier over the years of the study.

| Year | | Area 3D (ha/km²) | Area 2D (ha/km²) | Glacier front (m a.s.l.) | Area changes since 1981 (%) | Area changes since 1981 (% yr⁻¹) |
|---|---|---|---|---|---|---|
| 1981 | | 135.7/1.36 | 115.49/1.15 | 2,828 | – | – |
| 2011 | | 69.3/0.69 | 62.59/0.63 | 2,939 | −49.0 | −1.6 |
| 2020 | Principal | 43.97/0.44 | 47.8/0.48 | 3,011 | −61.7 | −1.6 |
| | Secondary | 3.82/0.38 | 4.2/0.04 | 3,170 | | |
| 2021 | Principal | 41.99/0.42 | 46.1/0.46 | 3,014 | −63.1 | −1.6 |
| | Secondary | 3.44/0.03 | 3.9/0.04 | 3,170 | | |
| 2022 | Principal | 38.29/0.38 | 44.6/0.45 | 3,026 | −64.7 | −1.6 |
| | Secondary | 2.9/0.03 | 3.52/0.03 | 3,170 | | |

Line 219-221. Again, not the most standard approach. Because mountain glaciers usually shrink in area as they thin, it is customary to use the mean area of the period to convert the volume loss into mean thickness change (e.g.: Berthier and others, 2004; Falaschi and others, 2022).

A: This point led us to interesting discussions. We agree that glaciers generally shrink in area as they thin. As in a recent previous work carried out in the Pyrenees (Vidaller et al., 2021), we intend here to determine the mean ice thickness loss considered only changes within the most recent surface. For example, for the period 1981-2022, we had only considered ice thickness loss within 2022 surface, to avoid overestimations. If we considered the glacier extent at the beginning (or mean extent) of these periods, we would take into account ice loss in areas where no ice was present during the whole period when calculating mean ice thickness.

Table 1. I would place this table in supplementary material and include an area, or even better a cumulative area change plot in Figure 2.

A: Amended as suggested as Table S5. Also, a cumulative area change plot in Figure 2.

[Figure]

*Figure 2: Appearance of the Aneto Glacier during the study period. (A) Photo (Fernando Biarge, Fototeca DPH) corresponding to Aneto Glacier in 1982. (B) Photo corresponding to Aneto Glacier in 2022. The red stars refer to the same location in both photos. (C) Map showing differences in the area of the glacier during the study period; the purple line delineates the extent of the glacier in 1981, the green line in 2011, and the orange line in*

*2020. The shading of the terrain was calculated from the 2011 LiDAR. The yellow triangle represents the summit of Aneto peak. (D) Cumulative area change plot of the Aneto Glacier for the years 1981 (purple), 2011 (green) and 2022 (orange).*

Figure 2. B) I would invert the position or the year labels on top so that they follow the glacier recession as represented in the map (2022 2011 1981). There appears to be a moraine in front of the glacier, probably a LIA feature marking the glacier extent. With a slight offset to the west of the represented area, you could include the entire area encompassed by this moraine and put in even larger perspective the recent area loss.

A: Yes, that is the LIA moraine and since it appears clearly in Figure 2C that covers a larger area than Figure 2B. The glacier extent lost since the LIA is well documented in the Pyrenees in Rico et al. (2017) and summarized in lines 415-417 as follows:

*"Although this work focuses on the period 1981-2022, the glaciers of the Aneto-Maladeta massif had about 610 ha at the end of the LIA, so they lost about 338 ha from 1850 to 1984 (Rico et al., 2017)."*

Line 265. I made a general comment regarding the different footprints of your data. If you use uniform grid base for your data this sentence could simply read: "Between 1981 and 2022 the mean ice thickness loss was 30.51 m". Also, if you use mean area (see comment of Line 219-221), you can remove the line between brackets.

A: Instead of using raster data, which would require either filling gaps in areas without observations or reducing the size of raster cells to achieve the same resolution as coarser observations, we compared 3D point clouds directly with CloudCompare's M2C3 tool. In addition, as we calculate ice thickness differences within the smallest surface (the most recent surface of the study period in each case), we prefer to keep the information in parentheses. Therefore, the readers will be aware about ice thickness differences considering the biggest glacier surface (the oldest surface of the study period in each case) of each period are taken into account. Also, this information was show in Table S6 in *Supplementary Material* as:

**Table S6:** Glacier thickness change over the year of the study.

| Method of calculation | 1981-2022 (m / m yr$^{-1}$) | 1981-2011 (m / m yr$^{-1}$) | 2011-2022 (m / m yr$^{-1}$) | 2020-2021 (m) | 2021-2022 (m) |
|---|---|---|---|---|---|
| Slope-perpendicular | -30.5 / -0.7 | -17.8 / -0.6 | -12.6 / -1.1 | -1.5 | -2.7 |
| Height change | -45.3 / -1.1 | -26.5 / -0.9 | -18.6 / -1.7 | -2.2 | -4.8 |

Line 275-277. This line could be removed if you include a mass balance time series graph in Fig. 3.

A: We preferred to keep this sentence and not include the mass balance time series graph because it could span over periods of varying duration, which would hinder realistic identification of temporal trends. Therefore, we haven't made any change in this regard.

Figure 3. Your color scale is not very good here. I think you should use a single color ramp, with linear intervals, from white to dark (0 to -80 m). B) I would recommend a longitudinal profile from the headwall to the glacier front, rather than a frequency distribution.

A: The change in B) was included as suggested. In contrast, we did not change the colour scale in A) since we prefer want to keep the same colour scale as in previous work (Vidaller et al., 2021). In this way, the colour scale represents in white the zone without changes, red colours when the differences in thickness go from -5 to -7 m, and a colour ramp from red to dark purple when the differences are among -7 to -80 m.

[Figure]

*"Figure 3: (A) Thickness loss of Aneto Glacier from 1981 to 2022. In the upper map, the black line delineates the glacier in 2022 while grey line represents the glacier in 1981. The arrow indicates the north direction (see Supplementary Material Figure S3 the maps for each period of the UAV surveys). (B) Distribution of thickness loss considering elevation bands (mean of each band) of 20 m."*

Figure 4. A) Again colorscale. You have a linear variable from 0 to 45 m. The most appropriate approach is a single color, linear, color ramp. By using several colors, you make the interpretation harder. Two colors would be useful if you had positive and negative values, which is not the case here. B) Again consider using a longitudinal profile here. Note you would share a distance axis and could even combine both plots in a single graph.

A: Similar to Figure 3, we retained the colour scale, this time with different colours, due to in Figure 3 we show ice thickness loss, meanwhile in Figure 4 we represent remaining ice thickness. This colour scale is informative in terms of ice accumulation and corresponds to snow depth cartographies in mountain areas (Revuelto et al., 2014). We consider that using multiple colours helps potential readers to easily identify areas with different ice depths. Regarding the use of longitudinal profile, we think that, given the shape of the glacier with much shorter distances to the east than to the west, that type of representation could be a source of confusion. In any case, and considering recommendations of RC2, Figure 4B is now changed to a boxplot that shows mean ice thickness loss in each altitudinal band. We have also included a representative GPR profile in *Supplementary Material* (Figure S5) to support this figure.

[Figure]

*"Figure 4: Ice thickness of Aneto Glacier in 2020. In map (A), the blue colour represents the zones of lesser ice thickness that are about to disappear, in contrast to the purple colours that represent the greatest ice thickness. The secondary body of the Aneto Glacier is coloured grey because no data are available for this glacier body and therefore no interpolation is possible. The boxplot (B) shows the mean glacier thickness in 2020 for each elevation band (20 m). A GPR profile is show in Supplementary Material as example of longitudinal radargram (SE-NW) of the glacier (Figure S5 in Supplementary Material)."*

[Figure]

*"Figure S5: Radargram 1062, representative of the western area. The radargram is represented from SE (0 m) to NW (1000 m), so, from the high part to the lower part of the glacier."*

Figure 5. Consider longitudinal profile here too. A stack of the three thickness profiles is the simplest way of showing the absolute and relative thinning. See comment in Fig. 4 regarding colorscale.

A: See previous comment about the appropriateness of a longitudinal profile. We have completed this Figure by adding a plot that represents the mean ice thickness loss for each altitudinal band for 1981, 2022 and 2022 years to better compare the changes in ice thickness among the three studied years (now Figure 5D).

[Figure]

*"Figure 5: Reconstruction of the ice thickness of Aneto Glacier at different times during the study period. (A) shows the thickness in 1981, (B) shows the thickness in 2011, and (C) shows the thickness in 2022. The blue colour represents the zones of lower ice thickness that are about to disappear, in contrast to the red colours that represent the greatest ice thickness. The secondary body of the Aneto Glacier is coloured grey because no data are available for this glacier body and therefore no interpolation is possible. (D) Comparison of the thickness of Aneto Glacier in 1981, 2011 and 2022, structures in elevation bands of 20 m."*

Line 318. Remove "as is well known" and consider including a reference for this statement.

A: Amended as suggested. Added reference: Palacios et al. (2022). Now the sentence is shown as (L372-373):

*"Glaciers erode the surface beneath the ice mass so that the subglacial topography is not a flat surface (Palacios et al, 2020)."*

Line 324-328. Note how your first sentence is part of methods and the rest discussion or methdos. This paragraph of the results should start in line 328.

A: Amended as suggested. These lines have been moved to the methods and adapted to the text.

Line 336. "The rate of area loss was uniform over time". Maybe use an average here? Also note the number of figures between lines 336 and 343. A multiple line plot could do the job here.

A: Following this recommendation, we now report the average value of area loss as follows:

*"(−2.2 ha yr−1, −2.2 ha yr−1, −2.1 ha yr−1, and −2.1 ha yr−1 from 1981 to 2011, 2020, 2021, and 2022, respectively)"* → *"(-2.2 ha yr⁻¹)"*

Since the total number of figures in the manuscript and the supplementary material is already 10, we prefer including these numerical values in the text instead a new plot, as they equally provide information about the observed temporal trends.

Lines 348-537. Consider a synthesis plot here (i.e.: Fig. 3 in Dussaillant and others, 2019), which allows the comparison of several datasets with different observational periods.

A: In these lines we wanted to discuss the specific glacier mass balance values observed in the Pyrenees with different techniques and time periods compared to the specific mass balance of the Aneto Glacier. Synthesis plots such as the one presented by Dussaillant et al. (2019) for the Andes, would be useful in a "regional" Pyrenean mass balance analysis but not necessary for a detailed study on a single glacier.

References

Berthier E, Arnaud Y, Baratoux D, Vincent C and Rémy F (2004) Recent rapid thinning of the "Mer de Glace" glacier derived from satellite optical images. Geophysical Research Letters 31(17). doi:10.1029/2004GL020706.

Dussaillant I and others (2019) Two decades of glacier mass loss along the Andes. Nature Geoscience 12, 802–808. doi:10.1038/s41561-019-0432-5.

Falaschi D and others (2022) Increased mass loss of glaciers in Volcán Domuyo (Argentinian Andes) between 1962 and 2020, revealed by aerial photos and satellite stereo imagery. Journal of Glaciology, 1–17. doi:10.1017/jog.2022.43.

WGMS (2009) Attribute description. World Glacier Monitoring Service, Zurich, Switzerland.

Citation: https://doi.org/10.5194/tc-2022-261-RC1

---

## Author Comment (AC2)

Dear Editor, dear Reviewers:

We are pleased to submit a revised version of our manuscript entitled "The Aneto Glacier (Central Pyrenees) evolution from 1981 to 2022: ice loss observed from historic aerial image photogrammetry and remote sensing techniques" by Vidaller et al where we have incorporated your suggestions and corrections. We thank you very much for the valuable comments that certainly helped to improve our manuscript. Please, note that we have also simplified the title by removing the term "recent" as this was not informative nor accurate enough.

Below we provide a point-by-point response to all comments raised in the reviews. We hope that the revised manuscript now meets the quality standards of *The Cryosphere*. I will be happy to answer any question you might have regarding this study.

We look forward to hearing from you.

Sincerely,

Ixeia Vidaller and co-authors

*Review of 'The Aneto Glacier (Central Pyrenees) evolution from 1981 to 2022: ice loss observed from historic aerial image photogrammetry and recent remote sensing techniques' submitted by Vidaller et al for publication in The Cryosphere*

In this manuscript, the authors examine the historic and recent geometric changes to Aneto Glacier, the largest remaining glacier in the Central Pyrenees. They leverage a variety of topographic data sources to measure area and thickness changes since 1981, with a focus on the last 3 years due to the availability of UAV surveys. They also conducted field GPR measurements of ice thickness to map the contemporary bedrock and reconstruct the current and historic ice thickness. They then use the former to assess bedrock topographic variability and possible overdeepenings.

In general, the assemblage of field measurements and historic surveys provides a rich overview of changes to Aneto Glacier, and the documentation of the demise of an iconic Pyrenean glacier will definitely be a worthwhile contribution to the literature. However, I have some major concerns relating to several aspects of the manuscript, all of which should be addressed before the manuscript will be suitable for publication.

1. Research aim. The research aims need to be reformulated somewhat. I also do not think the authors have addressed their aims in the discussion: e.g. anticipating the evolution of other temperate mountain glaciers (L70) or determining the inflexion point in glacier mass loss (L85). Meanwhile 'showing' the speed of glacier change is not really a scientific objective.

A: Thank you very much for your appreciation. It is true that the main objectives were incompletely addressed in the previous manuscript version, so we have reformulated and grouped them.

We totally support that, given the final stage that the Aneto Glacier is in, determining whether the rate of loss is accelerating (because there is feedback processes) or slowing (because the remaining ice is increasingly confined to the most favourable areas) is of inherent scientific interest and can be further extrapolated to other mountain regions that will be in a similar situation in the coming decades.

Now the objectives are described as follows (L93-100):

> *"This study aims analysing the recent evolution of the highest and largest glacier of the Pyrenees, the Aneto Glacier, by using the longest temporal dataset of glacier thickness loss in the Pyrenees. In addition, this study permits to assess the impact of a single extreme warm ablation season (2022) on glacier evolution. Due to the very last stage in which*

*Aneto glacier is, we report thickness and ice surface losses of this glacier from 1981 to 2022, to discern if the speed of loss accelerates (because the existence of feedback processes) or slow down (because the remaining ice is progressively restricted to the most favourable areas), that has an inherent scientific interest and may be extrapolated to other mountain areas that will face a similar situation in the coming decades. The evidence for the demise of Pyrenean glaciers in the coming decades using the Aneto Glacier as an iconic example is also of interest to highlight the dramatic consequences of rapid climate change in mountain areas."*

And in the discussion as (L397-406):

*"Thus, there has been no recent acceleration in surface loss per year, but the relative surface loss has increased. Oppositely, the rates of change in glacier thickness have increased during the study period ($-0.6$ m yr$^{-1}$ from 1981 to 2011 and -1.1 from 2011 to 2022), indicating an acceleration of glacier thinning, especially in the last decade and more pronounced in the last three years. In terms of specific mass balance, the losses are 0.6 m w.e. yr$-1$ for the period 1981–2022, 0.5 m w.e. yr$-1$ for the period 1981–2011, 1 m w.e. yr$-1$ for the period 2011–2022, 1.2 m w.e. yr$-1$ for the period 2020–2021, and 2.7 m w.e. yr$-1$ for the period 2021–2022. Based on these results, two inflexion points can be identified, one after 2011 and another after 2020, in both cases the thickness loss has accelerated sharply.".*

And in as (L479-480):

*"This behaviour could also anticipate the behaviour of other temperate mountain glaciers in their final deglaciation phases."*

2. Clarity of presentation of results. The authors refer to thickness change, but in some cases their meaning is not entirely clear or consistent with the literature. I believe that they are determining the slope-perpendicular surface change (as normally derived from the M3C2 algorithm in CloudCompare), but this differs strongly from the typical surface height change values reported from geodetic studies (e.g. Hugonnet et al., 2021) and the authors need to be clear about that difference, as well as to check that it is what they wish to present. I would recommend also presenting traditional thinning values as well as the total volumetric change of the glacier for each period of interest. This feeds into the need to improve the discussion/justification for considering surface areas vs planimetric areas (again, I recommend reporting both values), which are useful for distinct analyses. More importantly, in some cases the authors seem to equate mean changes in ice thickness change (ie thinning) with changes to the mean ice thickness (e.g. L265-269, L389-

392); these quantities differ fundamentally due to changes in glacier area due to thinning, and the authors need to reconsider these comparisons.

A: In this case, as in Vidaller et al. (2021), we considered that it is worthwhile to use the M3C2 algorithm when working with 3D point clouds in very steep terrain (Aneto Glacier slope in 2020 was 24.3º). Otherwise, the surface height changes (vertical height differences) are not very informative in these particular cases. As glacier slope increases due to glacier front retreat (López-Moreno et al., 2016), vertical height changes do not show the true ice loss.

To make this problem clearer, the text has been changed as follows (L245-248):

> *"Glacier thickness loss (normal surface differences, see Supplementary Material for more information) between these point clouds were computed using CloudCompare's M3C2 tool (James, 2017) to determine the differences (surface perpendicular) between the glacier surfaces observed in different years. Glacier change statistics were derived from this later comparison, calculated over the most recent (and smallest) glacier surface."*

The information on normal surface differences is included in the *Supplementary Material* as:

> **2.3 Glacier area outline, point cloud geolocation and glacier thickness loss computation**
>
> *In this study, ice thickness loss (perpendicular to the glacier surface) was computed using CloudCompare's M3C2 tool. This method was also used in Vidaller et al. (2021) to determine true reduction in ice thickness (no change in ice depth, which is by definition a vertical difference). This method is not the standard one used for comparison of glacier reduction when working over larger areas and with larger glaciers, where vertical changes are normally calculated Hugonnet et al. (2021).*
>
> *In order to compute height change values, the local slope of glacier surface was considered to determine the vertical changes as follows:*
>
> $$H = \frac{h}{\cos \propto}$$
>
> *Where H is the height change value, h is the ice thickness loss (slope-perpendicular) and α is the slope."*

To make the results of this study comparable with those obtained by the traditional method (height surface change) in other study areas, we also calculated the height surface change of Aneto Glacier for the period 1981-2011, 2011-2022 and 1981-2022, as recommended by RC2. The main results are shown in Table S6 and Figure S3 in the *Supplementary Material*. Doing that, our study

provides both a modified methodology, more adequate when working very small glaciers with high slopes, and, simultaneously, we present the more "traditional" data on ice depth to compare with previous datasets.

**Table S6:** Glacier thickness change over the year of the study.

| Method of calculation | 1981-2022 (m / m yr$^{-1}$) | 1981-2011 (m / m yr$^{-1}$) | 2011-2022 (m / m yr$^{-1}$) | 2020-2021 (m) | 2021-2022 (m) |
|---|---|---|---|---|---|
| Slope-perpendicular | -30.5 / -0.7 | -17.8 / -0.6 | -12.6 / -1.1 | -1.5 | -2.7 |
| Height change | -45.3 / -1.1 | -26.5 / -0.9 | -18.6 / -1.7 | -2.2 | -4.8 |

[Figure]

*"Figure S3: Thickness loss for the periods 2020-2021 (left) and 2021-2022 (right). Data acquired with UAVs surveys. Black arrow determined North direction. The extent of left map corresponds with 2021 Aneto Glacier surface, and the right map with the surface of 2022."*

Summarizing, since Aneto Glacier is a very small glacier, we considered it is better to calculate the 3D extent of the glacier. Since the slope of the glacier is not negligible (24.3º in 2020), the projected 2D surface would not accurately describe the surface changes since the slope also changed during the study period. Nonetheless, Table S5 now shows both the 3D and 2D (projected) surfaces.

**Table S5:** Main characteristics of the Aneto Glacier over the years of the study.

| Year | | Area 3D (ha/km$^2$) | Area 2D (ha/km$^2$) | Glacier front (m a.s.l.) | Area changes since 1981 (%) | Area changes since 1981 (% yr$^{-1}$) |
|---|---|---|---|---|---|---|
| 1981 | | 135.7/1.36 | 115.49/1.15 | 2,828 | – | – |
| 2011 | | 69.3/0.69 | 62.59/0.63 | 2,939 | −49.0 | −1.6 |
| 2020 | Principal | 43.97/0.44 | 47.8/0.48 | 3,011 | −61.7 | −1.6 |

| | | | | | | |
|---|---|---|---|---|---|---|
| | Secondary | 3.82/0.38 | 4.2/0.04 | 3,170 | | |
| 2021 | Principal | 41.99/0.42 | 46.1/0.46 | 3,014 | −63.1 | −1.6 |
| | Secondary | 3.44/0.03 | 3.9/0.04 | 3,170 | | |
| 2022 | Principal | 38.29/0.38 | 44.6/0.45 | 3,026 | −64.7 | −1.6 |

This computation is also described in detail in the manuscript as follows:

*"2.3 Glacier area outline, point cloud geolocation and glacier thickness loss computation:*

*Due to the small extent of these glaciers, the slope was considered in the calculation of glacier area to obtain the true glacier area (3D surface) rather than the 2D projection of glacier extent. This calculation is justified because the glaciers are strongly bound to wall cirques and these have a steepness of 24.3º in 2020. When the slope is not taken into account, the glacier surface is underestimated (Vidaller et al., 2021). Otherwise the 2D area computation would also be affected by the changes in slope during the study period."*

Finally, following the recommendation of RC2, we have clarified in the text that we are computing with M3C2 ice thickness loss and not mean ice thickness changes. As we said later in a minor comment, in this revised version we use the term ice thickness loss (not anymore ice thickness changes to avoid misunderstandings). This idea is described in the main manuscript in section 2.3: as follows:

*"Glacier thickness loss (normal surface differences, see Supplementary Material for more information) between these point clouds were determined using the CloudCompare tool M3C2 (James, 2017), to determine the differences (surface perpendicular) between the glacier surfaces observed in different years. Glacier change statistics were derived from this later comparison, calculated over the most recent (and smallest) glacier surface."*

And in the *Supplementary Material* as:

*"2.3 Glacier area outline, point cloud geolocation and glacier thickness loss computation*

*In this study, changes in ice thickness (perpendicular to the glacier surface) were computed using CloudCompare's M3C2 tool. This method was also used in Vidaller et al. (2021) to determine true reduction in ice thickness (no change in ice depth, which is by definition a vertical difference). This method is not the standard one used for comparison of glacier reduction when working over larger areas and with larger glaciers, where vertical changes are normally calculated (Hugonnet et al. (2021).*

*In order to compute height change values, the local slope of glacier surface was considered to determine the vertical changes as follows:*

$$H = \frac{h}{cos \propto}$$

*Where H is the height change value, h is the ice thickness loss (slope-perpendicular) and α is the slope value."*

3. Gaps in presentation of the results. The authors have compiled visually aesthetic figures, in general, but I think there are some fundamental gaps in their presentation of results. I would specifically request that they present the GPR transects in the main manuscript, along with an assessment of the overall uncertainty of their bedrock estimates (partial uncertainty from the crossing points in Table S2, but also consider the radar uncertainty in general), provide thinning maps for each period of the UAV surveys, include off glacier measured elevation changes (Figure 3 appears cropped) and an empirical uncertainty assessment of the thinning rates, which is necessary as the authors do not seem to have used independent GCPs to assess the accuracy of their UAV-derived elevation models. I also think that the thinning/volume change results would benefit from a temporal summary plot (also depicting uncertainties), as in e.g. (Ragettli, Bolch, & Pelliccotti, 2016).

A: Thank you for your appreciation of the figures. As we respond in a minor comment from RC2 below, the GPR transects in Figure S1 are included in the *Supplementary Material*, because there are a large number of transects (32) in a very small area (44 ha), so if we include the location of these transects in the interpolation, the ice thickness information is partially obscured.

Following the RC2 recommendation on GPR uncertainty, Table S2 is included in the *Supplementary Material*, to show GPR thickness differences. This table shows the intersections between nearly "perpendicular" transects where ice thickness differences were included as GPR uncertainty. To determine georadar uncertainty we estimated different velocities for temperate ice in three transects. This has been included in the method section (L272-277):

*"General GPR uncertainty in ice thicknesses was determined considering different velocities for temperate ice in the transects (1043, 1062 and 1073). Based on existing literature (Jimenez, 2016; López-Moreno et al., 2018), we assumed 0.2 m/ns in the snow and between 0.157 and 0.186 m/ns in the ice. With these velocities, mean and maximum ice thickness were determined for each transect (Table S3 in Supplementary Material). As a result, mean ice thickness variation that could be derived from different velocity into the*

*temperate ice would fit in the range of the estimated error band (< 1.6 m) and smaller than the uncertainties obtained from the differences in thickness at transect crossings (< 1.8 m)."*

More information about this dataset is included in the Supplementary Material in methods Table S3:

**Table S3:** Mean and maximum ice and snow thickness determined from the different velocities considered within the range of temperate ice. Zsmax acronym corresponds to maximum snow thickness, Zimax to the maximum ice thickness, Zsavg to the mean snow thickness and Ziavg to the mean ice thickness for each transect.

| Transect | Thickness | Vn=0.2 m/ns; Vh=0,163 m/ns | Vn=0.2 m/ns; Vh=0,168 m/ns | Vn=0.2 m/ns; Vh=0,157 m/ns |
|---|---|---|---|---|
| 1043 | Zsmax (m) | 5.59 | 5.59 | 5.59 |
|  | Zimax (m) | 11.95 | 12.32 | 11.51 |
|  | Zsavg (m) | 2.92 | 2.92 | 2.92 |
|  | Ziavg (m) | 7.18 | 7.40 | 6.92 |
| 1062 | Zsmax (m) | 2.37 | 2.37 | 2.37 |
|  | Zimax (m) | 32.26 | 33.25 | 31.07 |
|  | Zsavg (m) | 1.52 | 1.52 | 1.52 |
|  | Ziavg (m) | 12.80 | 13.19 | 12.33 |
| 1073 | Zsmax (m) | 4.53 | 4.53 | 4.53 |
|  | Zimax (m) | 31.35 | 32.31 | 30.20 |
|  | Zsavg (m) | 2.56 | 2.56 | 2.56 |
|  | Ziavg (m) | 20.53 | 21.16 | 19.78 |

RC2 asks for thinning maps for each period of the UAV surveys. In this case, we have added this figure is *Supplementary Material* as Figure S3. In addition, the UAV observations will be made available in a public repository (Zenodo). In the case of change of elevation out of glaciers, in Figure 3 is represented ice thickness loss per elevation band (slope-perpendicular) for the period 1981-2022, considering changes in 1981 surface, now delineated in grey. We consider, as the error is expressed in the manuscript no show changes out of glacier, due to there are near cero. In Vidaller et al. (2021) Figure S5, shows the elevation changes out of the glacier in the comparison 2011-2020 data, so this error could be extrapolated to the other comparison between LiDAR and UAVs data. This fact is mentioned in the revised manuscript.

Attending to the empirical uncertainty assessment of the thinning rates, the images obtained with the UAV allowed an accurate geolocation due to the PPK GPS positioning. We check that the point clouds were coregistered with each other in CloudCompare with several areas of stable terrain around the glaciers. In all areas the comparison between these surfaces was nearly zero (0.02 m). In the case of the comparison between UAV point clouds and LiDAR point clouds, we have followed the methodology of Vidaller et al. (2021), where the uncertainty for Aneto Glacier ice thickness loss along the 2011-2020 period was 0.63 m.

We agree with RC2 that a summary plot could be appropriate to group the main results of thinning, but we think that the number of figures presented in this study is already sufficient to correctly understand and represent the presented information. Moreover, differences in these time periods are included in Table S5 and S6 in Supplementary Material, furthermore described in the text.

4. Demonstration of the suitability of TPI for identifying bedrock overdeepenings. This is an interesting idea as an alternative to topographic-routing approaches to identify sinks, but the authors need to demonstrate that it is useful for this purpose – does this approach work to identify bedrock sinks in mountainous terrain? Analytically-speaking, I am concerned that, as TPI is also dependent on local topographic high values, the sinks that the authors identify are actually due to the neighbouring high topography of glacier headwalls, rather than locally low values of bedrock topography. The authors would also do well to compare to a conventional drainage-based identification of topographic lows.

A: Thank you for this comment and for suggesting the review of the suitability of TPI for identifying bedrock thresholds and depressions. We do not think that the presence of an abrupt topography at the headwall will promote the detection of a false sink, especially when this is detected when TPI is calculated for various search distances. This fact is also corroborated by the visual inspection of longitudinal GPR profiles that clearly show the existence of an overdeepening area at the site indicated by TPI maps. The literature review shows, TPI is able to identify terrain sinks at different search distances (Weiss, 2001). With this in mind, the text was modified as follows (L256-262):

> *"Topographic Position Index (TPI) is capable of identifying terrain depressions at various search distances (Weiss et al., 2001). From this basal topography, the TPI (de Reu et al., 2013) was derived for 70, 100, 150 and 200 m search distances to describe bedrock depression areas that potentially favour future lake formation. This index has been used previously in studies of debris-covered glaciers (Westoby et al., 2020) to determine areas of potential debris accumulation, but as far as the authors are aware, this is the first time this index has been used to determine areas of potential lake formation following the retreat of mountain glaciers. In addition, overdeepenings detected by TPI were corroborated using the longitudinal GPR radargrams."*

A good example of these depressions of the underlying glacier bedrock is radargram 1062, which crosses the glacier from NE to SW (Figure S5). In this radargram, there is a depression from 0 to 200 m with a depth greater than 20 m that matches the depression shown by TPI. The next

depression is deeper than 20 m in the radargram (600-1000 m) and it is reflected with high TPI values below the Maldito peak. This argument was added to the results (L385-387) as:

*"This spatial distribution of the lowest value of TPI is confirmed by radargram 1062 (Figure S5 in the Supplementary Material), in which the left side coincides with the overdeepening area below del Medio pass and also with the second depression below Maldito peak."*

[Figure]

*"Figure S5: Radargram 1062, representative of the western area. The radargram is represented from SE (0 m) to NW (1000 m), so, from the high part to the lower part of the glacier."*

For this reason, we believe that the basal topography observed with the GPR confirms the suitability of using TPI to determine possible sinks that could be future mountain lakes. Finally, we would like to point out the negative TPI values found for 150 and 200 m search distances near Innominato lake. Although this is a new lake (without GPR observations), fed by meltwater and still largely dammed by the glacier itself, the TPI values found in the glaciated area close to the lake, suggest a depression where the lake could expand.

5. Discussion. The discussion could be improved in several ways, and ideally should provide new insights. At present, the comparison to past studies (both Vidaller et al, 2021 and Campos et al, 2021) does not provide a clear, consistent explanation for discrepancies between values reported in past studies (note also my point 2 above relating to potential discrepancies between the physical variables measured, as well as the lacking rationale for surface area instead of planimetric area). This should all be clarified, but more importantly the Discussion should be geared to critically evaluate their own results and to provide new insights. The authors make a very casual assertion of albedo change and melt season duration prolongation, but without evidence, and attribute the

accelerated mass losses to warming, without presenting analysis of a climatic record. I also miss a critical evaluation of the possibility of lakes based on the TPI results. Most importantly, though, I feel the study misses a chance to postulate on the future changes of this system. As select examples: at the current rates of volume loss, how long do we expect Aneto to last? When would the hypothesized overdeepenings become exposed? Is the headwall likely to provide a microclimate that can prolong the glacier's life? Or, indeed, does Aneto still qualify as a glacier – is it moving dynamically? – and when may it be too small to be considered as such? Although the field data they have collected is commendable, I feel that the authors need to go further examining some of these aspects if they wish this study to be a meaningful contribution to the literature.

A: Thank you for the detailed explanation of the points to be improved in the discussion. The first point is the comparison with more recent studies. Thus, in the case of comparison with Vidaller et al. (2021) and, since there are no differences in the methods, this new study represents a temporal extension (2011-2020 from Vidaller et al. (2021) to 1981-2022 from this study). In the case of the study presented by Campos et al. (2021), as we argue in the discussion, there are differences in the area of the 1981 glacier whether they work with planimetric area (2D) or with surface area (3D). We have reviewed the two values of our study (L440-443).

> *"In 1983, they reported an area of 103.2 ha (1.03 km$^2$), in contrast to the 135.7 ha (1.36 km$^2$) for 1981 described in this work. The large difference may be due in part to the fact that they did not consider the slope angle of the terrain in their calculations (2D vs 3D surface). Nonetheless, considering our delineation, but ignoring the effect of slope angle on the area estimate, we would have reported a value of 115.5 ha (1.16 km$^2$) for 1981, which underestimates our value by 20%."*

Additionally, Campos et al. (2021) indicate that the extension of the glacier during 1983 should be considered with caution (L446-447).

> *"The extension for the 1983 stage should be considered with caution, due to the lower quality of the 1983 aerial Image."*

On the other hand, RC2 is concern about the fact that we make claims about albedo changes, but we have not directly measured it. Instead, we base our statement on visually observed albedo differences in fieldwork campaigns year after year. To improve this statement, we added new references and acknowledge the need of new detailed studies on albedo (L481-485):

> *"However, a detailed quantification of the darkening of the glacier surface and its effect on the energy and mass balance has not yet been carried out. Early spring/summer snowmelt and glacier thickness loss result in a grey/dark appearance of the glacier surface,*

*which reduces the albedo effect and increases the absorption of thermal energy, leading to an acceleration of glacier surface and thickness loss (Shaw et al., 2021)."*

Regarding the temperature increase recently observed, we based our statement on AEMET data of Renclusa station and general statistics data showed by AEMET in a report of the 2022 summer temperatures (L469-472):

*"On the contrary, we have observed the high vulnerability of the Aneto Glacier to the occurrence of extremely hot summers in recent years, as 2022, when summer temperatures were 0.5ºC above the mean for the period 2007-2022, according to Renclusa station (2,140 m a.s.l.) and almost 2ºC in general in the Iberian Peninsula (AEMET); thus, the continued losses of surface area and thickness could be due to an increase in temperature."*

Referring to TPI to evaluate the possibility of new lakes we think that TPI is a good method to detect potential depressions in the bedrock topography, even if a headwall is present and even if it could overestimate the depth of the overdeepenings. Others methods as drainage models, may be induce errors when the basal DEM is created by interpolating and extrapolating information from radargrams, as to obtain reliable drainage models, and high-quality drainage models are required. To make sure that TPI is a good method to detect overdeepenings, we have calculated it with difference search distances, and also, we have corroborated this overdeependings with the radargrams. This is now expressed in the methods section in the main manuscript as (L256-261):

*"Topographic Position Index (TPI) capable of identify terrain depressions at various search distances (Weiss et al., 2001). From this basal topography, the TPI (de Reu et al., 2013) was derived for 70, 100, 150 and 200 m search distances to describe depression areas that potentially favour future lake formation. This index has been used previously in studies of debris-covered glaciers (Westoby et al., 2020) to determine areas of potential debris accumulation, but as far as the authors are aware, this is the first time this index has been used to determine areas of potential lake formation following the retreat of mountain glaciers. In addition, overdeepenings detected by TPI were corroborated using the longitudinal GPR radargrams."*

And in results as (L385-387):

*"This spatial distribution of the lowest value of TPI is confirmed by radargram 1062 (Figure S5 in the Supplementary Material), in which the left side coincides with the overdeepening area below del Medio pass and also with the second depression below Maldito peak."*

Finally, the future change of these systems is a recurrent question in the scientific field and especially in the popular science world. We have discussed this point in the discussion as follows (L511-520):

*"How long the glacier will maintain the ice movement and a surface greater than 2ha to still be considered as a glacier is a very uncertain issue to be estimated. The duration of the glacier depends on several factors, such as the temperature evolution in the next few years, the evolution of precipitation (mainly snowfall in winter), the ability of the glacier to transport the debris fallen from the headwalls (and avoid the darkening of the surface), possible events of dust deposition (which may be frequent in winter and spring) and many other factors. In addition, according to the study by Vidaller et al. (2021), it is possible that these very small glaciers, once they become smaller than 10 ha, will have a greater topoclimatic control, so their preservation could be prolonged if there are no more very hot summers, as in 2022. Otherwise, glacier extinction could be imminent if there are few summers like 2022 in the next decade. However, more detailed studies are needed to answer such a simple question, to reduce the uncertainty of observations, simulations and also to provide a deeper understanding of those processes that govern small and very small glaciers."*

Minor comments:

L21. Please check 'glaciated' vs 'glacierized'

A: Amended as suggested

L39. (Huss & Hock, 2018) is a great modelling study but is not 'observed'

A: Deleted citation as all our data and comparisons are observed data. Thank you for the appreciation.

L44. Misplaced colon :

A: Changed by a comma.

L48. I suggest refraining for using 'ice thickness wastage', which is a vague term, and instead using volume loss or thinning, here and throughout the manuscript. Similarly, for clarity I would recommend 'area loss' instead of shrinkage.

A: There is an interesting point raised throughout this review, that has already led to an interesting discussion between the authors of the manuscript: the terminology, thinning vs wastage and surface loss vs shrinkage; as in previous works in the Pyrenees (Vidaller et al., 2021). We finally decided to use thickness and surface losses. We chose 'thickness loss' since our results refer to thickness, not volume. And in the case of 'shrinkage', we have removed "shrinkage" in references to area loss, but we referred these loss as 'surface loss' to make clear that we are working with the surface area (3D), due to the importance of the slope (>20°) in this very small glacier. To this end, we calculated mean ice thickness loss and glacier surface reduction (surface loss) using 3D surface and point clouds generated with structure-from-motion techniques.

L49. Can you be more specific about which volume change data is lacking? Clearly there is now the (Hugonnet et al., 2021) dataset, but perhaps this is not sufficiently precise or long-term?

A: Sentence rephrased as:

*"In terms of ice thickness wastage, unlike area shrinkage loss, there is in general a lack of information. Recent studies identified an ice thickness loss of 6.3 m for the period 2011–2020 as the mean for all the glaciers in the Pyrenean massif (Vidaller et al., 2021). Specifically, at Monte Perdido Glacier, López-Moreno et al. (2019) reported wastage of 6.1 m for the period 2011–2017. In the case of Ossoue Glacier, the geodetic mass balance was −31.3 ± 1.9 m w.e. (water equivalent) for the period 1983–2013 and −17.3 ± 2.9 m w.e. for the period 2001–2013 (Marti et al., 2015)."* → *"In terms of ice thickness loss, unlike surface loss, there is generally a lack of information over a long period of time and with a sufficient resolution for small alpine glaciers (or very small glaciers). Recent studies identified an ice thickness loss of 6.3 m for the period 2011–2020 as the mean for all the glaciers in the Pyrenean massif (Vidaller et al., 2021). Specifically, at Monte Perdido Glacier, López-Moreno et al. (2019) reported ice thickness loss of 6.1 m for the period 2011–2017. In the case of the Ossoue Glacier, the ice thickness loss was −36.8 m for the period 1983–2013 and −20.4 m for the period 2001–2013 (Marti et al., 2015). In the grid cell corresponding to the Pyrenean glaciers (1°×1° grids 42°N, 0°E and 42°N, 1°W) of, Hugonnet et al. (2021) indicated a mean ice thinning rate of -0.96 m yr-1 for the period 2000-2019, which is very accurate considering the dataset characteristics, but it is much higher than the mean annual ice thickness loss found by Vidaller et al., 2021 of −0.70 m*

*yr−1, for a more recent study period ( higher ice loses could be expected in this later period). This difference between both studies clearly shows the need of local studies as the present study or Vidaller et al., 2021 to validate large scale observations and also to reach more accurate estimations over shorter time periods".*

L52-53. Please refer to thinning rates for this comparison.

A: Amended as suggested.

*"In the case of Ossoue Glacier, the geodetic mass balance −31.3 ± 1.9 m w.e. (water equivalent) for the period 1983–2013 and −17.3 ± 2.9 m w.e. for the period 2001–2013 (Marti et al., 2015)."* → *"In the case of the Ossoue Glacier, the ice thickness loss was −36.8 m for the period 1983–2013 and 20.4 m for the period 2001–2013 (Marti et al., 2015)."*

L55. 'considered' is redundant here

A: Amended as suggested.

L59. 'having an additional protection figure' – unclear what you mean here – I suppose that the authors refer to additional societal value?

A: Yes, we refer to an additional societal value. Rephrased as:

*"...having an additional protection figures for this natural landscape heritage..."* → *"...having an additional societal value for this natural landscape heritage…"*

L66. 'comprehension' -> analysis

A: Amended as suggested.

L67. Is this the largest spatio-temporal dataset of glacier observations? I don't think that is necessary motivation for this study

A: We meant dataset of thickness loss in the Pyrenees. Of course, there is not the main motivation for this study, but there is another point to look at this study.

> *"This study aims to exploit the largest spatio-temporal dataset of glacier observations to analyse…"* → *"This study aims analysing the recent evolution of the highest and largest glacier of the Pyrenees, the Aneto Glacier, by using the longest temporal dataset of glacier thickness loss in the Pyrenees…"*

L68-71. Please consider the scientific objectives: 'showing' consequences is not a valid scientific motivation (implies bias), but investigating/measuring/understanding is.

A: You are right, thank you. "show" has been replaced by "understand".

L72. Please use either 'subject of annual monitoring' or 'subjected to annual monitoring'

A: Amended as suggested.

L91. I do not understand what is meant by 'sectors can be delineated…'

A: We agree with you; this part of the sentence does not correspond to the aim of our study and is out of context. The sentence has been deleted in final manuscript version.

L94. 'longest period ever observed' seems like hyperbole – have there been investigations of the LIA extent? LGM?

A: There are no other previous studies in this area that cover longer periods of time. Nevertheless, to soften the sentence, we have deleted the word "ever".

L108. Is this the annual isotherm or a seasonal isotherm?

A: This is the mean annual isotherm. It has been corrected.

L110. This formulation of the temperatures is not very clear to me (L112-115 for the Aneto summit is much clearer). Over what period are these the maximum and minimum mean annual temperatures?

A: We have reformulated this paragraph and describe only the data for Renclusa station, since they cover a longer period of time, and we have express data as:

*"Mean annual temperature for the period 2007-2022 was 4.6 ℃ at the weather station of Renclusa hut (2,140 m a.s.l.), meanwhile the mean temperature for the same period in the ablation season (June-September) was 11.6 ℃. 2022 was an especially warm year, in which annual mean temperature was 5.2 ℃ and the summer mean temperature was 12.1 ℃ (data from the AEMET database)."*

L141 Please check how to acknowledge this source in the 'Acknowledgements' section of the manuscript. Can you ascertain any information with regards to the precision of geospatial positioning of the data?

A: Following your recommendation, we have acknowledged the "QGIS service 'QuickMapServices'" and other institutions in the 'Funding' section as follows:

*"We thank the Spanish National Geographic Institute (IGN) for the collection, archiving and distribution of the aerial photographs. We also thank the NextGIS/QuickMapServices plugin (Original Work Published in 2014), available online at https://github.com/nextgis/quickmapservices (accessed June 2, 2021). And to AEMET to share the climatic data of Renclusa hut."*

We also changed the sentence (L146-147) in the methods as follows:

*"For this study, the historical aerial imagery was digitalized at a resolution of 15 microns."*
→ *"For this study, the historical aerial imagery was rescanned at a high resolution of 15 microns."*

Regarding your question on the precision, we have added (L 161-164):

*"the georeferencing accuracy of DigitalGlobe's latest Very High Resolution (VHR) satellites (i.e. GeoEye-1 and WorldView-1/2/3/4) ranges from 1.0 to 5.0 m, which can be insufficient for many precise geodetic applications. To improve this, we aligned the 1981 point cloud with that of 2020 using the ICP algorithm (Rajendra et al., 2014)."*

L142. Do you use any independent check points?

A: We did not use independent check points to improve the accuracy of the 1981 point cloud. However, we chose to match this point cloud with the 2020 UAV-derived point cloud, which is very accurate after PPK processing, as explained in the text (L. 239-243):

> *"To coregister the LiDAR point cloud (2011) and the point cloud from the historical aerial imagery (1981), several areas of stable terrain such as ridges, peaks, polished surfaces, etc. were selected in these later point clouds and in the 2020 UAV-derived point cloud. These areas were evenly distributed around the glacier. A rotation and translation matrix was calculated for these areas to align (separately) the 1981 and 2011 point clouds with that of 2020 using an ICP algorithm (Rajendra et al., 2014), from Cloud Compare software (Girardeau-Montaut, 2016), in the same way as Vidaller et al. (2021)."*

L159. How does using the same protocol imply reduced uncertainties? I can imagine the argument that it implies similar uncertainties to a prior study, but this should be borne out with direct measurements.

A: We agree with RC2 that using the same protocol is not necessary a way to reduce uncertainties. To avoid misunderstanding, we have deleted the last part of the sentence (L182 now) and indicated that we urge to always use the same protocol in order to compare different studies.

L162-3. Please rephrase this statement – have you then assessed the distribution of deviations on stable ground? Is this from individual check points or the distribution of absolute deviations, or?

A: RC2 is correct, for this study we did not use check points or assessed the distribution of deviations on stable ground, but in other studies such as Revuelto et al. (2021) (this reference was added to the main manuscript) they confirm the feasibility of comparing UAV point clouds. Also, we have added the elevation variation out of glacier for the comparison between 2021-2022 DEMs, and referred in the main manuscript as (L184-186):

> *"Due to the three UAV acquisitions having the same acquisition protocol, and the GPS-PPK geolocation (images geolocation with deviations below 4 cm), the comparison of these three point clouds yielded negligible deviations (0.06 m) (Revuelto et al., 2021)."*

L192. Have these data been submitted to GlaThiDa (Welty et al., 2020)?

A: Not yet, but we can consider submitting at the time of the publication of the manuscript, thank you.

L199. (also 237-239) How was the division done? Is this totally random, or patch-wise? Note that your radar returns are nearly continuous and autocorrelated, so a random subsetting is likely to give a very good cross-validation result, but has little to no information with regards to the accuracy of your interpolated product.

A: The division was made arbitrarily. We agree with you that the cross-validation gives very good results because of the distribution of the data along the transects, but we have no stacks on the glacier to validate it using an additional method.

L207. Strong slope = steep. Please give a slope value?

A: Amended as suggested and added the slope value (24.3°).

> *"This calculation is justified because the glaciers are strongly bound to wall cirques and these have a steepness of 24.3° in 2020."*

L211. Is this the 3D or xy geolocation accuracy? I disagree that using the same protocol will give the same results – this is dependent on weather conditions (especially wind and humidity) as well as the stability of camera focus (for example).

A: It is an XYZ geolocation accuracy. We mean that it has the same order of magnitude for the three UAV acquisition. We understand that the thickness of the glacier changes, but no Z or elevation out of the glacier. You are correct that the results could be different with the same protocol, but we compared the elevation changes in the stable terrain part of the 2021 and 2022 point clouds, and the differences are very close to 0. In any case, we have added the mean XYZ RMS error of the geolocation for the three UAV acquisitions as (L235-238):

> *"This geolocation error is equivalent for the 2020, 2021, and 2022 point clouds (0.019 for 2020, 0.025 for 2021, and 0.021 for 2022, the differences were due to weather conditions). Based on the low magnitude of these geolocation errors, we assume that the error introduced in ice thickness differences is nearly negligible. 3D point cloud differences in*

*ice free areas had RMSE below 0.02 m, (error computed following Vidaller et al., 2021 accuracy method)"*

L217. M3C2 gives the surface-perpendicular distance, rather than thinning. Does it also give a total volume-change estimate?

A: Yes, M3C2 gives the perpendicular distance of the surface, so we assume it is ice thickness loss and no vertical differences. M3C2 does not calculate the volume change.

L222. Please note the uncertainty of this assumed value.

A: Amended as suggested, adding uncertainty of density conversion factor ($\pm60$ kg/m$^3$).

*"The mass balance was calculated assuming a density conversion factor of 850±60 kg/m3 (Huss, 2013)."*

L225. The TPI method is an interesting idea, but it is not very clearly justified – in particular because TPI values are extremely low directly beneath steep areas, whether or not the location is a topographic low. Is there a justification for using TPI instead of a drainage model, as is commonly used (Linsbauer, Paul, & Haeberli, 2012)?

A: As we have already explained, we think that TPI is a good method to detect potential depressions in the bedrock topography, even if a headwall is present and even if it could overestimate the depth of the overdeepenings. In addition, we are not sure how well a drainage model works when the basal DEM is created by interpolating and extrapolating information from radargrams, as to obtain reliable drainage models, high quality drainage models are required.

L240-242. Do you have an estimate of the uncertainty of delineation from the 1981 survey?

A: Yes, we do. As mentioned in the 'Data and methods' section 2.3 of the manuscript, the glacier delineation from the 1981 survey was done manually using the same procedure as in Vidaller et al. (2021). As with the orthomosaics derived from UAV flights in 2020, 2021, and 2022, and LiDAR survey in 2011, the uncertainty of the 1981 orthomosaic was determined as the root of the quadratic sum of four different sources of error (Rabatel et al., 2011).

For 1981 survey the uncertainty was computed taking into account four error parameters (see Vidaller et al., 2021): (1) error from the pixel size of the high-resolution orthomosaic we obtained from photogrammetric processing (±0.4 m), (2) error due to the geometric correction corresponded to 3.17 pixels (±1.27 m) according to the root mean squared error (RMSE) from the absolute geolocation variance taken from the Agisoft Metashape processing report, (3) error in the delineation was defined as 5 pixels (±2 m) larger than in Rabatel et al. (2011) and Vidaller et al. (2021) due to small differences during the digitization process by two operators, and (4) error due to marginal snow cover was set to 10 pixels (±4 m) due to residual snow cover in some parts of the glacier margin. The final uncertainty of 1981 survey the total uncertainty was the sum of the root of the quadratic sum of four different sources of error (Rabatel et al., 2021).

The uncertainty error of the 1981 glacier outline is 0.58 ha (0.0058 km$^2$). This represents an uncertainty error of 0.43 % of the total glacier area, added in the main manuscript as (L282-285):

*"The surface uncertainty is 0.048 ha (0.00048 km$^2$) for Aneto Glacier (Vidaller et al., 2021) in the case of the glacier surface of 2011, 2020, 2021 and 2022; the uncertainty error of the 1981 glacier outline is 0.58 ha (0.0058 km$^2$)."*

However, to clarify this part, we have added the following paragraph in section 2.4 of the *Supplementary Material*:

*"In this study, using the same procedure as in Vidaller et al. (2021), the uncertainty of the glacier outlines was determined as the root of the quadratic sum of four different sources of error and multiplied by the perimeter of the glacier outline, as previously described by Rabatel et al. (2011). Table S4 lists the errors for each year and the resulting uncertainties of each orthomosaic."*

*"Table S4: Details of the errors associated with the orthomosaics produced for this study. The largest error is associated with geometric correction and residual snow cover in 1981. All images were rectified based on 2020 UAV point cloud."*

| Year | Photo/Image source | Scale/Pixel size | Error due to the pixel size (m) | Error due to the geometric correction (m) | Error in the delineation (m) | Error due to marginal snow cover (m) | Total uncertainty (m) |
|---|---|---|---|---|---|---|---|
| 1981 | IGN | 0.35 m | 0.4 | 1.27 | 2 | 4 | 2.8 |
| 2020 | UAV | 0.03 m | 0.1 | 0.02 | 0.2 | 0.3 | 0.79 |
| 2021 | UAV | 0.03 m | 0.1 | 0.02 | 0.2 | 0.2 | 0.72 |
| 2022 | UAV | 0.03 m | 0.1 | 0.02 | 0.2 | 0 | 0.57 |

L245. 300% larger than?

A: The test of coregistration of point clouds from historical aerial imagery, LiDAR and UAV was performed considering a buffer of 300% of the extent of the 1981 surface of Aneto Glacier, so we can be sure that this area was never covered by ice in the period 1981-2022.

> *"This means that the comparison of the 1981 and 2020 point clouds was performed in a buffer zone with a 300% larger extent around the 1981 glacier boundaries (over stable terrain)"* → *"This means that the comparison of the 1981 and 2020 point clouds was performed in a buffer zone with a 300% larger extent than the 1981 glacier boundaries (over stable terrain)".*

L251. By 'no movement' you seem to mean that the terminus is not retreating higher, is that correct? Or did you assess surface velocity?

A: By 'no movement' we mean surface velocity. This secondary body is decreasing its surface area from year to year like the principal body.

> *"It is remarkable that the secondary body has no movement today"* → *"It is noteworthy that the secondary body today shows signs of stagnant dynamics".*

Table 1. Please consider providing uncertainty estimates for your area-change assessments.

A: Uncertainty estimates are described in the methods section, and we believe that it is not necessary to report uncertainties for all area change results because they are nearly redundant.

L265. Does this 'mean ice thickness loss' correspond to the thinning (i.e volume change over the glacier area) or the change in the mean ice thickness over time? These are two very different properties that need to be clearly disambiguated (both are worth noting!).

A: The mean ice thickness loss corresponds to the mean change in ice surface over the period considered. This has been clarified in the text as follows (L311-315):

> *"A comparison of the 1981 and 2022 point clouds (difference calculated normal to surface) shows a mean ice thickness loss of 30.51 m (Figure 3 and Figure S3 and S4 in Supplementary Material) in this period and considering only the area occupied by the glacier in 2022 (considering the 1981 glacier extent, the ice thickness loss is 24.1 m; and considering height surface changes the losses are 45.3 m (for more information see Table*

*S6 and Figure S4 in Supplementary Material)). Note these mean ice thicknesses loss, are the mean value of differences in glacier surfaces (normally computed) for the entire period computed."*

[Figure]

*"Figure S4: Map A) represents ice height differences (considering differences in the vertical plane) for the period for the period 1981-2011. The thickness (and outer) boundary represents 1981 Aneto Glacier surface, meanwhile the inner black line 2011 Aneto Glacier surface. Map B) shows the ice height differences for the period for the period 2011-2022. The thickness (and outer) boundary represents 2011 Aneto Glacier surface, meanwhile the inner black line 2022 Aneto Glacier surface. Map C) corresponds to ice height differences for the period for the whole period (1981-2022). The thickness (and outer) boundary represents 1981 Aneto Glacier surface, meanwhile the inner black line 2022 Aneto Glacier surface. Map D) represents thickness variation (slope-perpendicular) for the period for the whole period (1981-2022). The thickness (and outer) boundary represents 1981 Aneto Glacier surface, meanwhile the inner black line 2022 Aneto Glacier surface. Black arrow represents North direction. The difference between the two methods show as in this case and due to the small size and high slope, the results of A), B) and C) maps are overestimated."*

L275. Please consider presenting volume changes (loss is a negative number).

We understand RC2's concerns about the sentence in L275. Therefore, we have rephrased that sentence:

> "As for the specific mass balance, the losses are 0.6 m w.e. $yr^{-1}$ for the period 1981–2022, 0.5 m w.e. $yr^{-1}$ for the period 1981–2011, 1.0 m w.e. $yr^{-1}$ for the period 2011–2022, 1.2 m w.e. $yr^{-1}$ for the period 2020–2021, and 2.7 m w.e. $yr^{-1}$ for the period 2021–2022." → "As for the specific mass balance, the changes are -0.6 m w.e. $yr^{-1}$ for the period 1981–2022, -0.5 m w.e. $yr^{-1}$ for the period 1981–2011, -1.0 m w.e. $yr^{-1}$ for the period 2011–2022, -1.2 m w.e. $yr^{-1}$ for the period 2020–2021, and -2.7 m w.e. $yr^{-1}$ for the period 2021–2022."

Figure 3. If I understand correctly, this figure presents the surface lowering (ice thinning). Please also indicate the extents of the glacier in 1981 and 2022 on the upper panel.

A: Yes, you are correct, as we explained in previous comments, we calculated the changes in thickness perpendicular to the surface (M3C2 tool). We changed the extent of the glacier in 2020 by the extent in 2022 (black) and we added the extent in 1981 (grey). Thank you very much for the appreciation.

[Figure]

*"Figure 3: (A) Thickness loss of Aneto Glacier from 1981 to 2022. In the upper map, the black line delineates the glacier in 2022 while grey line represents the glacier in 1981. The arrow indicates the north direction (see Supplementary Material Figure S3 the maps for each period of the UAV surveys). (B) Distribution of thickness loss considering elevation bands (mean of each band) of 20 m."*

Figure 4. Please indicate the locations of the GPR lines on this left panel. On the right panel, I would consider a box plot rather than the range of thicknesses (the dots), which will give a better sense of the distribution of thicknesses for each elevation.

A: The GRP lines are in the Supplementary Material because the important information (ice thickness) would be obscured if we included them in the aforementioned panel as we have already commented in a previous comment. According to this suggestion for improvement, we have replaced Figure 4B with a boxplot.

[Figure]

*"Figure 4: Ice thickness of Aneto Glacier in 2020. In map (A), the blue colour represents the zones of lesser ice thickness that are about to disappear, in contrast to the purple colours that represent the greatest ice thickness. The secondary body of the Aneto Glacier is coloured grey because no data are available for this glacier body and therefore no interpolation is possible. The boxplot (B) shows the mean glacier thickness in 2020 for each elevation band (20 m). A GPR profile is show in Supplementary Material as example of longitudinal radargram (SE-NW) of the glacier (Figure S5 in Supplementary Material)."*

L298-300. This is Methods.

A: Yes, you are right. These three lines are part of the methods, we have eliminated them.

L302-303. Glacier thickness was high -> glacier was thick

A: Amended as suggested.

L306. Please use past tense also for the 2022 observations.

A: Amended as suggested.

Figure 5. Panel A looks quite noisy due to localized artefacts in the 1981 dataset, which you could probably justify filtering out. Could you use these reconstructed geometries to look at the volume area relationship for this glacier (Bahr, Pfeffer, & Kaser, 2015)?

A: It is true that the 1981 map has some noise compared to the other maps. In this case, we have already applied a low filter to reduce the noise. We considered that applying more low filters or even high filters could change the actual information.

We consider that presenting only three points (dates) of volume-area relationship does not provide enough meaningful information about a topic that is not in the main scope of the manuscript.

L325. Misplaced period after the Westoby citation

A: This section was deleted at the suggestion of RC1.

L325-330. I think some more justification and demonstration for the TPI method is needed – can you validate the approach with now-exposed overdeepenings, for instance the Innominato Lake from the 1981 data? How does this compare to a flood-filled glacier bed DEM? How is this method better for identifying overdeepenings?

A: Again, thank you for the comment. As we showed in the 4th major comment, we already checked the identification of depressions showing TPI values with some of the radargrams.

If we understand correctly, you ask about the differences of a basal topography with overdeepening, and a basal topography that have these depressions filled with water. As we said before, TPI has been evaluated and compared with the radargrams results and, in the case of the overdeepenings, if they were filled by water, this change of material (ice-water-rock) would have to be reflected as variations in wave speed, so an extra layer in the radargram between the ice and the rock would be visible.

We consider that this method is the best one here due to it is able to detect thresholds and depressions with different search distances compared to the possible noise involved in a DEM produced with interpolated data from the GPR profiles.

Figure 6. The locus of low TPI values below the headwall is due to the headwall, rather than due to the subglacial topography.

A: We think that the low values below the headwalls are favoured by the headwalls, but there is really a depression there. Otherwise, the TPI values would not be negative, especially when considering different search distances. We showed before that this overdeepening is clearly visible from radargrams.

L344. Was there any accumulation area in 2021 or 2022?

A: In 2022, the ablation period was too long (compared to other years), from mid-June to mid-October; moreover, winter precipitation was very low. Therefore, at the end of the summer, the entire glacier was free of snow, with only a little snow near the bergschrund. In the case of 2021, summer temperatures were lower than 2021, and winter precipitation were higher, even so the accumulation area that year was very little and only near the bergschrund. So in both years, we can consider the accumulation area negligible.

L345. Neither PlanetScope nor Sentinel-2 are sufficiently high resolution?

A: The resolution of PlanetScope is 4 m and of Sentinel-2 10 m. In recent snowy years, no more than the 30 m higher part of the glacier is covered with snow, so these two satellites do not have high enough resolution.

L358-367. I feel that some of this discussion could simply be confusion about what a previous study reported, and how this calculation works. The mass balance is the mean value of thinning, not the change of the mean thickness. It is clear that the mean thickness will not decrease at the same rate as the surface thins, because the mean thickness is only calculated for the glacier area that is still glacier. Please reconsider this section. I would suggest to also refer to the Hugonnet et al (2021) results for this domain, even if they are imprecise due to resolution and so on. In fact, I would strongly recommend a figure compiling and comparing the mass balance observations available from this study, Vidaller et al (2021), Campo et al (2021) and Hugonnet et al (2021), as well as the observed spatial extents.

A: Thank you for the appreciation; the mass balance point could lead to misunderstanding. In this study, we always consider only the changes in the surface area where the glacier is still preserved,

both if we show the thickness loss or the specific mass balance. To clarify this issue, we have added some sentences in the text (L252-253):

*"Thus, the specific mass balance presented in this study was determined considering the recent surface of the glacier."*

Another example (L324-327):

*"As for the specific mass balance, the volume changes are -0.6 m w.e. yr−1 for the period 1981–2022, -0.5 m w.e. yr−1 for the period 1981–2011, -1.0 m w.e. yr−1 for the period 2011–2022, -1.2 m w.e. yr−1 for the period 2020–2021, and -2.7 m w.e. yr−1 for the period 2021–2022 (data always calculated within the most recent glacier surface)." (L402-403)*
*"In terms of specific mass balance (considering only changes at the smallest surface glacier, the most recent year of comparison), the losses are 0.6 m w.e. yr$^{-1}$…"*

RC2 also, suggest referring here to Huggonet et al (2021). We have included a reference to Hugonnet et al. (2021) as follows (L414-415):

*"Hugonnet et al. (2021) also determined a mean ice thinning of -0.96 m yr$^1$ for the Pyrenean glaciers for the period 2000-2019."*

Regarding the purpose of RC2 about a figure compiling and comparing the mass balance observations available from this study, Vidaller et al (2021), Campo et al (2021) and Hugonnet et al (2021), given the differences in the study period of each work such a figure could result confusing for readers. We prefer, however, mentioning and cited the results in the revised manuscript as it is now.

L374-382. I have a hard time understanding the authors' argument here – what is the advantage that they see in investigating surface area changes instead of planimetric changes? As there is a considerable deviation due to the high surface slopes, should the authors not simply report both?

A: Following R2 recommendation, and to make this study comparable to other studies, we also included 2D planimetric areas. We use the 3D surface instead of the 2D planimetric area because this glacier is a very small glacier that has a not negligible slope, so using the 2D area we underestimate the area, and this underestimation is larger the smaller the glacier and consequently its slope. In the discussion we partially associate the difference to 2D estimation reported by Campos et al. (2021); but the different quality of the scanning procedure might also explain part of the differences.

L388. Have the authors compared the spatial distribution of measured ice thicknesses from their own observations and Campos et al (2021)? I see clearly that there is a statistical difference but the authors need to demonstrate that this is not simply due to spatial biases in sampling. How do the observed bedrock elevations compare in space?

A: The spatial distribution of the GPR data from Campos et al. (2021) and our study is very similar, but the thickness values are very different. We also agree with the reviewer that comparing the maximum glacier thickness is not the best way to go because of interpolation issues, but information in Campos et al. (2021) is very limited and no furthers comparisons are possible. In our study, in the case of the 2020 data, we were careful with the maximum data, the maximum values that always came from GPR measurements, and never from interpolation data, which could be a misinterpretation.

L401. This link to hot summers in recent years has not been explicitly made, and would require a comparison of (best if long-term) measured mass balance along with climate information. Although this is very likely to be true, the authors have not tested or demonstrated this, only that the glacier is continuing to rapidly lose mass.

A: Thank you very much for your appreciation. It is true that we did not compare measured mass balance with climate data. We have changed the paragraph of climatic data in the *1.1 Study area* section as (L127-130):

> *"Mean annual temperature for the period 2007-2022 was 4.6 ºC at the weather station of Renclusa hut (2,140 m a.s.l.), meanwhile the mean temperature for the same period in the ablation season (June-September) was 11.6 ºC. 2022 was an especially warm year, in which mean annual temperature was 5.2 ºC and the mean summer temperature was 12.1 ºC (data from the AEMET database)."*

And in the *4.3 Future* perspectives as (L467-472):

> *"The rate of surface and thickness loss calculated in this study and the reconstruction of ice thickness for the year 2022 indicate the critical situation of this glacier. There are no signs of slowdown in glacier surface and thickness loss rates; on the contrary, we have observed the high vulnerability of the Aneto Glacier to the occurrence of extremely hot summers in recent years, as 2022, when summer temperatures were 0.5ºC above the mean for the period 2007-2022 (according to Renclusa station (2,140 m a.s.l.)); thus, the continued loss of surface area and thickness could be due to an increase in temperature."*

L409-414. Please write this more clearly, e.g. heat -> energy. Also, have you evaluated the albedo changes (not demonstrated here), or is this a hypothesis? Same for the longer exposure of the glacier – please clearly differentiate between observations and conceptual discussions.

A: We have replaced "heat" with "energy", and thank you for the appreciation.

We have not evaluated the changes in albedo of this glacier, it is simply a hypothesis based on observations in the field and the results of calculating glacier thickness and surface area losses in recent years. As noted in a previous comment above, this is clarified in the revised text as (L481-485):

*"However, a detailed quantification of the darkening of the glacier surface and its effect on the energy and mass balance has not yet been carried out. Early spring/summer snowmelt and glacier thickness loss result in a grey/dark appearance of the glacier surface, which reduces the albedo effect and increases the absorption of thermal energy, leading to an acceleration of glacier surface and thickness losses (Shaw et al., 2021)."*

L411-412, L424-426 The references to Shaw et al (), Otto (), and Yue et al () need to be reformatted.

A: The references have been reformatted.

L450. Realizing that this would entail a change of scope, I think it would be very worthwhile to also consider the future evolution of this glacier and its neighbours. See e.g (Huss & Fischer, 2016).

A: Thank you for your recommendation, but as you say, this would mean a change in scope. This future evolution, although not in an expanded form, is proposed in Vidaller et al. (2021) for the glaciers of the Pyrenees. As mentioned earlier, it is difficult to provide sound information on this regard.

L478. Alignment issue with this reference

A: Amended as suggested.

L470 – Several reference formatting issues. Please ensure that you follow the guide for The Cryosphere.

A: The references have been reformatted.

References:

Bahr, D. B., Pfeffer, W. T., & Kaser, G. (2015). A review of volume-area scaling of glaciers. Reviews of Geophysics, 53(1), 95–140. https://doi.org/10.1002/2014RG000470

Hugonnet, R., McNabb, R., Berthier, E., Menounos, B., Nuth, C., Girod, L., … Kääb, A. (2021). Accelerated global glacier mass loss in the early twenty-first century. Nature, 592(July 2020). https://doi.org/10.1038/s41586-021-03436-z

Huss, M., & Fischer, M. (2016). Sensitivity of very small glaciers in the swiss alps to future climate change. Frontiers in Earth Science, 4(April), 1–17. https://doi.org/10.3389/feart.2016.00034

Huss, M., & Hock, R. (2018). Global-scale hydrological response to future glacier mass loss. Nature Climate Change, 8(2), 135–140. https://doi.org/10.1038/s41558-017-0049-x

Linsbauer, A., Paul, F., & Haeberli, W. (2012). Modeling glacier thickness distribution and bed topography over entire mountain ranges with glabtop: Application of a fast and robust approach. Journal of Geophysical Research: Earth Surface, 117(3), 1–17. https://doi.org/10.1029/2011JF002313

Ragettli, S., Bolch, T., & Pellicciotti, F. (2016). Heterogeneous glacier thinning patterns over the last 40 years in Langtang Himal. The Cryosphere, 10(5), 2075–2097. https://doi.org/10.5194/tc2016-25

Welty, E., Zemp, M., Navarro, F., Huss, M., Fürst, J. J., Gärtner-Roer, I., … Li, H. (2020). Worldwide version-controlled database of glacier thickness observations. Earth System Science Data, 12(4), 3039–3055. https://doi.org/10.5194/essd-12-3039-2020

---

## Author Response (AR1)

[revised manuscript text omitted]

In this study, ice thickness loss (perpendicular to the glacier surface) was computed using CloudCompare's M3C2 tool. This method was also used in Vidaller et al. (2021) to determine true reduction in ice thickness (no change in ice depth, which is by definition a vertical difference). This method is not the standard one used for comparison of glacier reduction when working over larger areas and with larger glaciers, where vertical changes are normally calculated (Hugonnet et al., 2021).

In order to compute height change values, the local slope of glacier surface was considered to determine the vertical changes as follows:

$$H = \frac{h}{\cos \alpha}$$

Where H is the height change value, h is the ice thickness loss (slope-perpendicular) and α is the slope value.

**2.4 Correction and accuracy assessment**

[revised manuscript text omitted]

835    **Table S4:** Details of the errors associated with the orthomosaics produced for this study. The largest error is associated with geometric correction and residual snow cover in 1981. All images were rectified based on 2020 UAV point cloud. Using the same procedure as in Vidaller et al. (2021), the uncertainty of the glacier outlines was determined as the root of the quadratic sum of four different sources of error and multiplied by the perimeter of the glacier outline, as previously described by Rabatel et al. (2011).

| Year | Photo/Image source | Scale/Pixel size | Error due to the pixel size (m) | Error due to the geometric correction (m) | Error in the delineation (m) | Error due to marginal snow cover (m) | Total uncertainty (m) |
|---|---|---|---|---|---|---|---|
| 1981 | IGN | 0.35 m | 0.4 | 1.27 | 2 | 4 | 2.8 |
| 2020 | UAV | 0.03 m | 0.1 | 0.02 | 0.2 | 0.3 | 0.79 |
| 2021 | UAV | 0.03 m | 0.1 | 0.02 | 0.2 | 0.2 | 0.72 |
| 2022 | UAV | 0.03 m | 0.1 | 0.02 | 0.2 | 0 | 0.57 |

840    **Table S5:** Main characteristics of the Aneto Glacier over the years of the study.

| Year | | Area 3D (ha/km$^2$) | Area 2D (ha/km$^2$) | Glacier front (m a.s.l.) | Area changes since 1981 (%) | Area changes since 1981 (% yr$^{-1}$) |
|---|---|---|---|---|---|---|
| 1981 | | 135.7/1.36 | 115.49/1.15 | 2,828 | − | − |
| 2011 | | 69.3/0.69 | 62.59/0.63 | 2,939 | −49.0 | −1.6 |
| 2020 | Principal | 43.97/0.44 | 47.8/0.48 | 3,011 | −61.7 | −1.6 |
|  | Secondary | 3.82/0.38 | 4.2/0.04 | 3,170 |  |  |
| 2021 | Principal | 41.99/0.42 | 46.1/0.46 | 3,014 | −63.1 | −1.6 |
|  | Secondary | 3.44/0.03 | 3.9/0.04 | 3,170 |  |  |
| 2022 | Principal | 38.29/0.38 | 44.6/0.45 | 3,026 | −64.7 | −1.6 |

| | Secondary | 2.9/0.03 | 3.52/0.03 | 3,170 | | |

**Table S6:** Glacier thickness change over the year of the study.

| Method of calculation | 1981-2022 (m / m yr$^{-1}$) | 1981-2011 (m / m yr$^{-1}$) | 2011-2022 (m / m yr$^{-1}$) | 2020-2021 (m) | 2021-2022 (m) |
|---|---|---|---|---|---|
| Slope-perpendicular | -30.5 / -0.7 | -17.8 / -0.6 | -12.6 / -1.1 | -1.5 | -2.7 |
| Height change | -45.3 / -1.1 | -26.5 / -0.9 | -18.6 / -1.7 | -2.2 | -4.8 |

845

**Supplementary figures:**

[Figure]

**Figure S1:** Purple lines indicate radargram transects with their ID number (see the characteristics of each radargram in Table S1).

[Figure]

**Figure S2:** Radargram with the speed obtained in each diffraction hyperbole, considering the established RWV of snow and ice (0,200 and 0,163 m/ns respectively).

[Figure]

**Figure S3:** Thickness loss for the periods 2020-2021 (left) and 2021-2022 (right). Data acquired with UAVs surveys. Black arrow determined North direction. The extent of left map corresponds with 2021 Aneto Glacier surface, and the right map with the surface of 2022.

[Figure]

**Figure S4:** Map A) represents ice height differences (considering differences in the vertical plane) for the period for the period 1981-2011. The thickness (and outer) boundary represents 1981 Aneto Glacier surface, meanwhile the inner black line 2011 Aneto Glacier surface. Map B) shows the ice height differences for the period for the period 2011-2022. The thickness (and outer) boundary represents 2011 Aneto Glacier surface, meanwhile the inner black line 2022 Aneto Glacier surface. Map C) corresponds to ice height differences for the period for the whole period (1981-2022). The thickness (and outer) boundary represents 1981 Aneto Glacier surface, meanwhile the inner black line 2022 Aneto Glacier surface. Map D) represents thickness variation (slope-perpendicular) for the period for the whole period (1981-2022). The thickness (and outer) boundary represents 1981 Aneto Glacier surface, meanwhile the inner black line 2022 Aneto Glacier surface. Black arrow represents North direction. The difference between the two methods show as in this case and due to the small size and high slope, the results of A), B) and C) maps are overestimated.

[Figure]

**Figure S5:** Radargram 1062, representative of the western area. The radargram is represented from SE (0 m) to NW (1000 m), so, from the high part to the lower part of the glacier.

875